# Biodegradation of highly crystallized poly(ethylene terephthalate) through cell surface codisplay of bacterial PETase and hydrophobin

Zhuozhi Chen[1,4], Rongdi Duan[1,4], Yunjie Xiao[1,4], Yi Wei[1,4], Hanxiao Zhang[1], Xinzhao Sun[1], Shen Wang[1], Yingying Cheng[1], Xue Wang[1], Shanwei Tong[1], Yunxiao Yao [1], Cheng Zhu [1], Haitao Yang[1,2,3], Yanyan Wang [1] & Zefang Wang [1,2]

The process of recycling poly(ethylene terephthalate) (PET) remains a major challenge due to the enzymatic degradation of high-crystallinity PET (hcPET). Recently, a bacterial PET-degrading enzyme, PETase, was found to have the ability to degrade the hcPET, but with low enzymatic activity. Here we present an engineered whole-cell biocatalyst to simulate both the adsorption and degradation steps in the enzymatic degradation process of PETase to achieve the efficient degradation of hcPET. Our data shows that the adhesive unit hydrophobin and degradation unit PETase are functionally displayed on the surface of yeast cells. The turnover rate of the whole-cell biocatalyst toward hcPET (crystallinity of 45%) dramatically increases approximately 328.8-fold compared with that of purified PETase at 30 °C. In addition, molecular dynamics simulations explain how the enhanced adhesion can promote the enzymatic degradation of PET. This study demonstrates engineering the whole-cell catalyst is an efficient strategy for biodegradation of PET.

Poly(ethylene terephthalate) (PET) is among the most used plastics around the world[1,2]. However, large amounts of nondegradable PET waste has rapidly accumulated in ecosystems worldwide and has become a global issue[3]. Chemical, physical, and biological methods of recycling PET waste have been performed to counter this severe environmental problem[4,5]. Among these recycling methods, enzymatic hydrolysis of PET can provide a green route for recycling PET and exhibits several advantages, such as being environmentally friendly, saving energy, and minimizing waste[6–8]. In the last decade, a variety of PET hydrolases have been found, including esterases, lipases, and cutinases[9–12]. These enzymes can depolymerize amorphous or low-crystallinity PET to a certain extent. A recent landmark study showed that a genetically engineered LCC enzyme could achieve a minimum 90% depolymerization of the postconsumer colored-flake PET waste within 10 h at 72 °C;[9] however, most of the reported PET hydrolases can hardly degrade the high-crystallinity PET (hcPET) that is routinely used for manufacturing bottles and textiles at mild temaperatures[6,13].

In 2016, a research group from Keio University identified a novel PET-hydrolyzing enzyme named PETase from the bacterium *Ideonella sakaiensis* 201-F6[14]. One of the most important characteristics of

[1]School of Life Sciences, Tianjin Key Laboratory of Function and Application of Biological Macromolecular Structures, College of Precision Instrument and Opto-electronics Engineering, Key Laboratory of Systems Bioengineering (Ministry of Education), Tianjin University, Tianjin 300072, China. [2]Tianjin International Joint Academy of Biotechnology and Medicine, Tianjin 300457, China. [3]Shanghai Institute for Advanced Immunochemical Studies and School of Life Science and Technology, Shanghai Tech University, Shanghai 201210, China. [4]These authors contributed equally: Zhuozhi Chen, Rongdi Duan, Yunjie Xiao, Yi Wei. e-mail: yanyanwang@tju.edu.cn; zefangwang@tju.edu.cn

PETase is its ability to degrade hcPET at a low temperature (30 °C); due to this property, PETase is a promising candidate for a practical method of recycling PET. However, the industrial application of PETase faces a substantial problem, i.e., the enzymatic activity of PETase toward hcPET remains low. Some pioneering studies have been conducted to address this issue. Numerous enzyme mutants have been rationally designed based on the structure of PETase and exhibited an increase in enzymatic activity toward low-crystallinity PET (lcPET)[15–17]. Only a few PETase mutants can improve the degradation capacity of hcPET to some extent[9,18–22]. Although previous studies demonstrated that redesigning PETase was a promising route to enhance its enzyme performance, the current enzymatic activity of PETase and its variants remains insufficient to realize the industrial application of hcPET degradation.

How can the degradation efficiency of PETase be further increased? The enzymatic degradation of PET is a two-step process in which the enzyme binds to the polymer substrate and subsequently catalyzes hydrolytic cleavage[23,24]. So far, most studies on PETase have focused on the second catalytic step[15,16,18,19,22,25–27], and the possible impact of the adsorption step on the degradation efficiency of PETase has received little attention[19,28–33]. Recently, we constructed a whole-cell biocatalyst by displaying PETase on the surface of yeast cells[34]. Considering that the whole-cell biocatalyst showed certain advantages over purified PETase, we continued to optimize the surface display system to further increase its degradation efficiency toward hcPET. By analyzing the crystal structure of PETase, it was clear that PETase lacked apparent substrate binding motifs, such as the carbohydrate-binding modules, which are generally observed in glycoside hydrolases[15,16]. Several binding modules have been reported to enhance the enzymatic degradation of polymers by increasing enzyme adsorption[31,35]. Therefore, it is possible to increase the degradation efficiency of the whole-cell biocatalyst by introducing an external adhesive unit into the surface display system to control the process of PETase-displaying cells adsorbing on the PET surface.

Hydrophobins (HFBs), a type of small secreted protein produced by filamentous fungi, play different roles in fungal growth and development, such as the production and dispersion of spores, formation of aerial hyphae, and stabilization of fruiting body structures[36–39]. In addition to their roles in aerial growth and reproduction, HFBs can mediate fungal attachment to hydrophobic surfaces[40]. For example, hydrophobic conidiospores that are dispersed by wind or insects easily adhere to water-repellent biotic or abiotic substrates[41]. The SC3 hydrophobin is involved in attaching Schizophyllum commune hyphae to hydrophobic surfaces, such as Teflon. These interesting phenomena were attributed to the self-assembly of HFBs at the interface between the cell wall and the hydrophobic substrate[42]. During the self-assembly process, a hydrophobic patch on the hydrophobin surface could bind to hydrophobic surfaces through strong hydrophobic interactions[43–46]. Due to its unique amphipathic protein structure, hydrophobin plays a natural role in manipulating the cell surface hydrophobicity of filamentous fungi. Furthermore, hydrophobin has been demonstrated to increase the attachment of several enzymes (including PET-degrading enzymes) onto different substrates in vitro[47–49]. The speculated mechanism that underlies these observations is that hydrophobin can wet the surface of PET, and as a result, enzymes can more easily contact and attack the hydrophobin-modified PET surface[29,49,50]. Inspired by this appealing property of hydrophobin, we proposed that hydrophobin could be used to modify the surface hydrophobicity of PETase-displaying yeast cells to facilitate their attachment on the hydrophobic PET surface and ultimately enhance the degradation efficiency of the whole-cell biocatalyst. To test our idea, we developed a codisplay system by simultaneously displaying PETase and hydrophobin on the yeast surface. We used the class II hydrophobin I from *Trichoderma reesei* (HFBI) as an example, which is a small, amphiphilic globular protein that readily self-assembles at hydrophilic and hydrophobic interfaces[51,52].

In this work, the data confirms that both PETase and HFBI are functionally displayed on the yeast cell surface. The displayed HFBI can profoundly increase the hydrophobicity of the yeast cells; thus, as expected, the attachment of codisplayed cells onto the PET surface is improved. This codisplay system shows ~328.8-fold higher degradation efficiency than that of the native PETase toward the hcPET (crystallinity of 45%) at 30 °C. The corresponding conversion level for hcPET is ~10.9% (depolymerization rate of 20.92 $mg_{products}$ $d^{-1}$ $mg_{enzyme}^{-1}$) at 30 °C within 10 days. Our study provides a rational organization of different functional units on the microbial surface for enhanced biocatalytic activity, which could find more applications in biocatalysis, biosensing, and bioenergy.

## Results and discussion

### HFBI and PETase were functionally codisplayed on the surface of yeast cells

In our codisplay system, hydrophobin HFBI and PETase should play different roles based on their unique protein structures. HFBI is thought to regulate the adsorption of yeast cells on the substrate PET. PETase is responsible for degrading the substrate PET. Figure 1a shows the structures of the two functional proteins (PETase and HFBI) that are codisplayed on the yeast surface. The fuchsia part in the surface of HFBI represents a hydrophobic patch, which provides strong hydrophobic interactions and is how HFBI binds to various hydrophobic surfaces[53]. For PETase, the portion circled in the structure is the active site, which is much broader than those of the other PET hydrolases; this active site may explain why PETase can accommodate a large substrate, such as PET, at moderate temperatures[15,16]. Figure 1b shows a schematic diagram of our codisplay system. The anchoring proteins GCW51 and GCW61 were used to display PETase and HFBI on the yeast surface, respectively. A flexible linker (GGGGSGGGGS) was employed to link the corresponding anchoring protein with PETase and HFBI separately to prevent interference from occuring between each anchoring protein and its displaying target[54]. To explore the structural difference between wild-type PETase and linker-attached PETase, the crystal structures of those two proteins were determined at 2.0 Å (Fig. 1a, Supplementary Fig. 1a, and Supplementary Table 1) and 1.5 Å (Fig. 1c and Supplementary Table 1), respectively. We compared the structure of wild-type PETase and linker-attached PETase and found that the tertiary structures of these two proteins were quite similar, with an overall RMSD of 0.352 Å (Supplementary Fig. 1b). In the structure of linker-attached PETase, we can only see a part of the linker (6 amino acids) in the form of an irregular structure, indicating that the $(GGGGS)_2$ linker is flexible. This result is consistent with previous findings that the GGGGS sequence is the most widely used flexible linker that can appropriately separate the functional domains[55]. The catalytic centers of these two proteins were compared as well. As shown in Fig. 1d, the two active-site pockets were almost identical. The above results revealed that the C-terminal fused linker did not induce obvious structural changes in PETase. To validate our structural findings, we also performed molecular dynamics (MD) studies for the codisplayed PETase and HFBI (Fig. 1e and Supplementary Fig. 2). In the simulation, the $(GGGGS)_2$ linker showed high flexibility, and no unique structure was observed when the linker-attached PETase was displayed by the yeast cell-wall protein GCW51. Moreover, this flexible linker separated PETase from GCW51 in space, meaning that the two functional units did not affect each other.

Theoretically, our codisplay system should perform dual functions that mimic the two-step process of PET degradation[23,24]. The first function, which is derived from the HFBI part, is the ability to self-assemble and facilitate the binding of yeast cells onto the hydrophobic PET surface. The second function, which is inherited from PETase, is to enzymatically degrade the highly crystallized substrate PET.

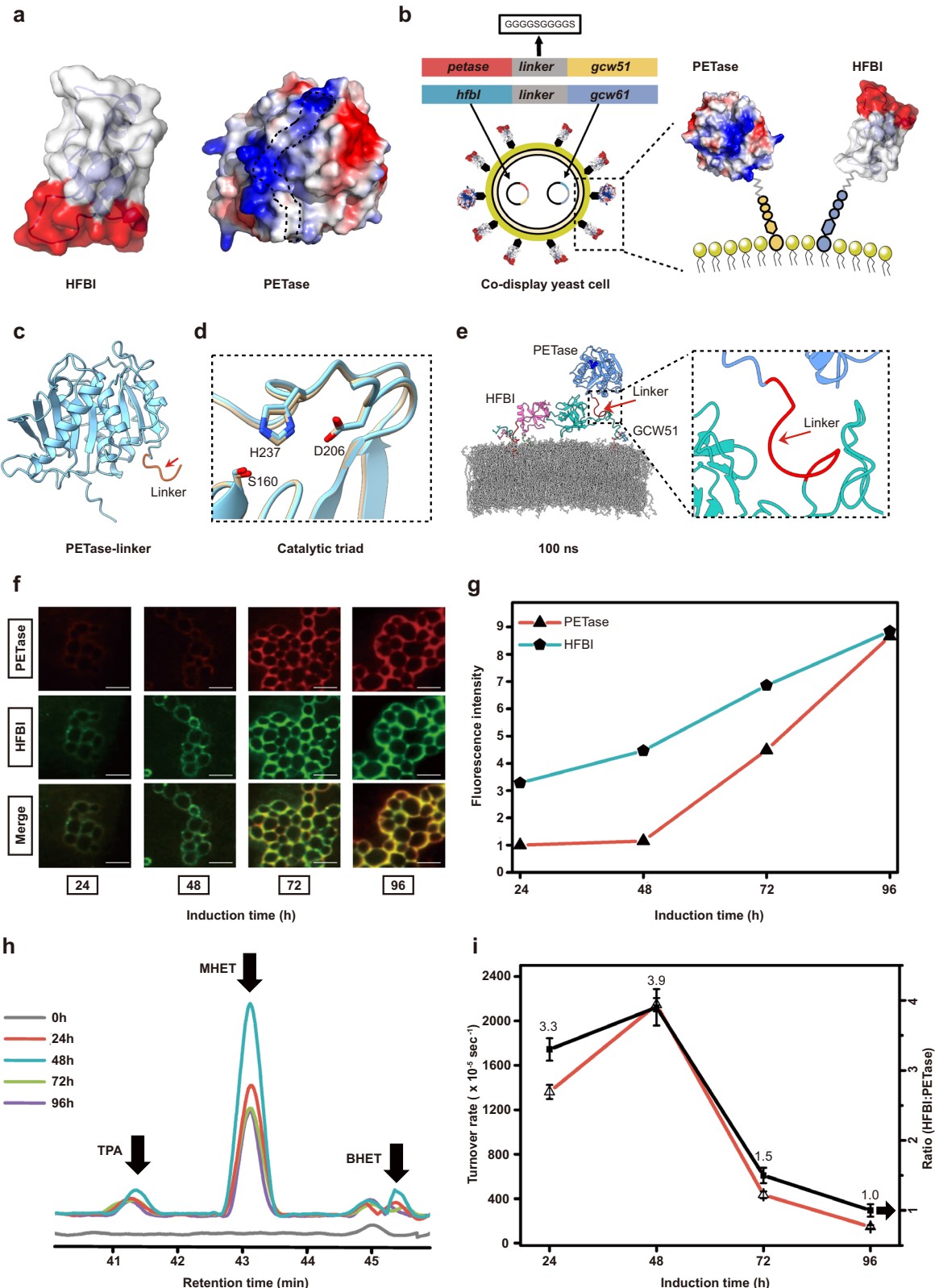

Supplementary Fig. 3 shows the Western blot (WB) results of both PETase and HFBI at different induction times. Positive bands on the WB film indicated that both proteins were successfully expressed in the recombinant yeast. To further demonstrate that HFBI and PETase were successfully displayed on the yeast cell surface, an immunolocalization assay was performed. A flag tag was introduced to the N-terminus of PETase as a response to the detection of anti-flag antibodies. For the displayed HFBI, a monoclonal antibody against HFBI was used in the immunofluorescence assay. As shown in Fig. 1f, strong fluorescent signals of PETase and HFBI were observed on the yeast cell surfaces. Additionally, no fluorescent signal was detected inside the cells. Furthermore, WB analysis of different cell fractions (whole cells, cell walls, and protoplasts) was performed to detect the cellular location of linker-attached PETase. As shown in Supplementary Fig. 4, immune

**Fig. 1 | Protein expression and function of *P. pastoris* GS115/PETase-HFBI at different induction times. a** X-ray crystal structure of the amphiphilic hydrophobin HFBI (the hydrophobic patch is marked in red) and PETase (blue indicates positive charge, red indicates negative charge, and the dotted line indicates active pocket position). **b** Schematic diagram of the codisplay system. **c** The overall structure of PETase-linker (the red arrow is marked as Linker). **d** Catalytic triad comparison of wild-type PETase with PETase-linker. **e** MD simulations of the codisplay system over 100 ns. Each domain is assigned a unique color. **f** Fluorescence microscopy of immunostained *P. pastoris* cells expressing PETase and HFBI on their surface under different induction times. Cells were labeled with a primary rabbit anti-FLAG antibody followed by a fluorescently labeled secondary goat anti-rabbit antibody and a primary mouse anti-HFBI antibody followed by a fluorescently labeled secondary goat anti-mouse antibody. The scale bar is 5 μm. The experiment is repeated three times independently, with similar results obtained. One representative is shown. **g** The intensity of the average fluorescence of tested cells was determined with ImageJ software. **h** HPLC analysis of the products released from the hcPET film degraded by PETase and HFBI codisplayed on the yeast cell. **i** Quantitative analysis of HPLC results and the protein expression ratio. The turnover rate was used to evaluate the enzyme activity of GS115/PETase-HFBI codisplay cells. $n = 3$ independent experiments. Data were presented as mean values ± SD. Source data for panels (**g**–**i**) are provided as a Source Data file.

signals of PETase mainly originated from the cell wall faction, apart from the whole-cell fraction. No obvious signals were detected from the protoplast fraction, regardless of the codisplay and display system. All the above results confirmed that both proteins were displayed on the yeast cell surface, as expected. In addition, the intensity of fluorescence was increased with increasing induction time, and the amount of surface-displayed HFBI was greater than that of the surface-displayed PETase when the induction time was less than 96 h (Fig. 1g). These results indicated that induction time played an important role in the levels of PETase and HFBI expression on the cell surface[34].

After we obtained the recombinant yeast cell that codisplayed PETase and HFBI, we were eager to know whether the displayed PETase maintained its enzymatic activity when it was codisplayed with hydrophobin HFBI. Figure 1h shows the degradation products released by the recombinant yeast cell that codisplayed PETase and HFBI by using hcPET as the substrate. The major product released by the displayed PETase was mono(2-hydroxyethyl) terephthalic acid (MHET), and trace amounts of terephthalic acid (TPA) and bis(2-hydroxyethyl)-TPA (BHET) were detected at the same time. The ratio between the liberated products of displayed PETase and that of the native PETase was almost the same (Supplementary Fig. 5d). Together, these results indicated that the displayed PETase was biologically active and exhibited the same reaction mechanism as that of native PETase, cleaving the polymer preferentially at similar positions[14,56].

The turnover rate of the recombinant yeast cell was calculated by quantitatively analyzing the high-performance liquid chromatography (HPLC) results from Fig. 1h. The standard curve of each released product was calculated to accurately measure the amount of each product (Supplementary Fig. 5a–c). A semiquantitative WB method was then used to estimate how much PETase was displayed in the yeast cells. We first generated a standard curve of wild-type PETase by performing grayscale analysis with the WB results (Supplementary Fig. 6a, b). Subsequently, yeast cells (at 48 h induction) were subjected to the same WB analysis to detect the displayed PETase (Supplementary Fig. 6c). According to the standard curve, 151.2 ng of PETase in a total of $2 \times 10^7$ displayed yeast cells. For the Codisplay system, $2 \times 10^7$ cells contained 35.2 ng of displayed PETase. As shown in Fig. 1i and Supplementary Fig. 7, the turnover rate of the recombinant yeast cell was dramatically affected by the induction time and the ratio between HFBI and PETase that was displayed on the cell surface. The ratio reached a maximum value at the induction time of 48 h when the ratio between HFBI and PETase displayed on the cell surface also reached its maximum. The above results revealed that the amount of HFBI on the cell surface exhibited an obvious impact on the enzymatic activity of the displayed PETase[57].

In our codisplay system, PETase is theoretically responsible for degrading the substrate PET. To confirm that surface-displayed HFBI does not degrade PET as well, three surface display systems were constructed, including the PETase (S160A)/HFBI codisplay system, linker-GCW51 (removing PETase)/HFBI codisplay system and HFBI single-display system, which are shown in Supplementary Fig. 8a-c. Enzymatic analysis of each surface display system revealed that none of the displayed systems constructed above exhibited PET degradation activity, as expected (Supplementary Fig. 8d, e), suggesting that PETase performed an enzymatic function rather than hydrophobin on the surface of yeast cells.

## Surface-displayed HFBI can increase the adsorption of codisplayed cells on PET surfaces by improving their surface hydrophobicity

In the above study, we confirmed the surface expression of hydrophobin HFBI. Microbial adhesion to hydrocarbons (MATH) and water contact angle (WCA) measurements were performed to verify whether the displayed HFBI could induce alterations in the surface hydrophobicity of the recombinant yeast cells. MATH is a classical technique for determining the surface hydrophobicity of various cells[58,59]. This method involves examining the adhesion of cells to liquid hydrocarbons (e.g., n-butanol, p-xylene), as this adhesion is directly associated with the hydrophobic surface properties of cells (Fig. 2a)[60]. As shown in Fig. 2b, all yeast cells in the control group (without induction) remained in the lower aqueous phase, indicating that the cell surface properties were hydrophilic. As induction time increased, the yeast cells that codisplayed PETase and HFBI gradually moved into the upper oil phase, causing a concomitant loss in turbidity throughout the lower aqueous phase. The relative hydrophobicity of the cell surface defined in the Methods section continued to increase with the induction time and reached a maximum at 96 h (Fig. 2c). To further demonstrate that the increase in cell surface hydrophobicity was mainly due to the surfaced-displayed HFBI, a MATH assay was conducted by using yeast cells that displayed only PETase. Supplementary Fig. 9 shows that yeast cells with only PETase remained in the lower aqueous phase within the entire induction time. Together, these results demonstrated that the displayed HFBI was responsible for the increase in the codisplayed cell hydrophobicity.

WCA is another commonly used method to measure the surface hydrophobicity of tested cells[61]. After performing high-density cell modifications, the WCA of the hcPET film can be determined by examining the cell surface properties. As shown in Fig. 2d, e, the hcPET film displayed a typical hydrophobic property (WCA is 85°). After modifications were performed with yeast cells that codisplayed PETase and HFBI (48 h induction), the WCA of the hcPET film slightly changed from 85° to 73°, indicating the codisplayed yeast cells exhibited a hydrophobic surface property. To clarify the contribution of the displayed HFBI to the hydrophobicity of the codisplayed yeast cells, the WCA of yeast cells displaying only PETase was also determined. The hydrophilic property (WCA is 37°) of the PETase-displayed cell confirmed that the HFBI part played an essential role in determining the cell surface hydrophobicity. Together, these results verified that by exposing its hydrophobic patch on the protein surface, the displayed HFBI can indeed increase the hydrophobicity of the codisplayed cells[43–46].

To determine whether the effect of displayed HFBI on cell hydrophobicity can be transformed into an effect on the attachment of yeast cells on the PET surface, we observed the adsorption of the

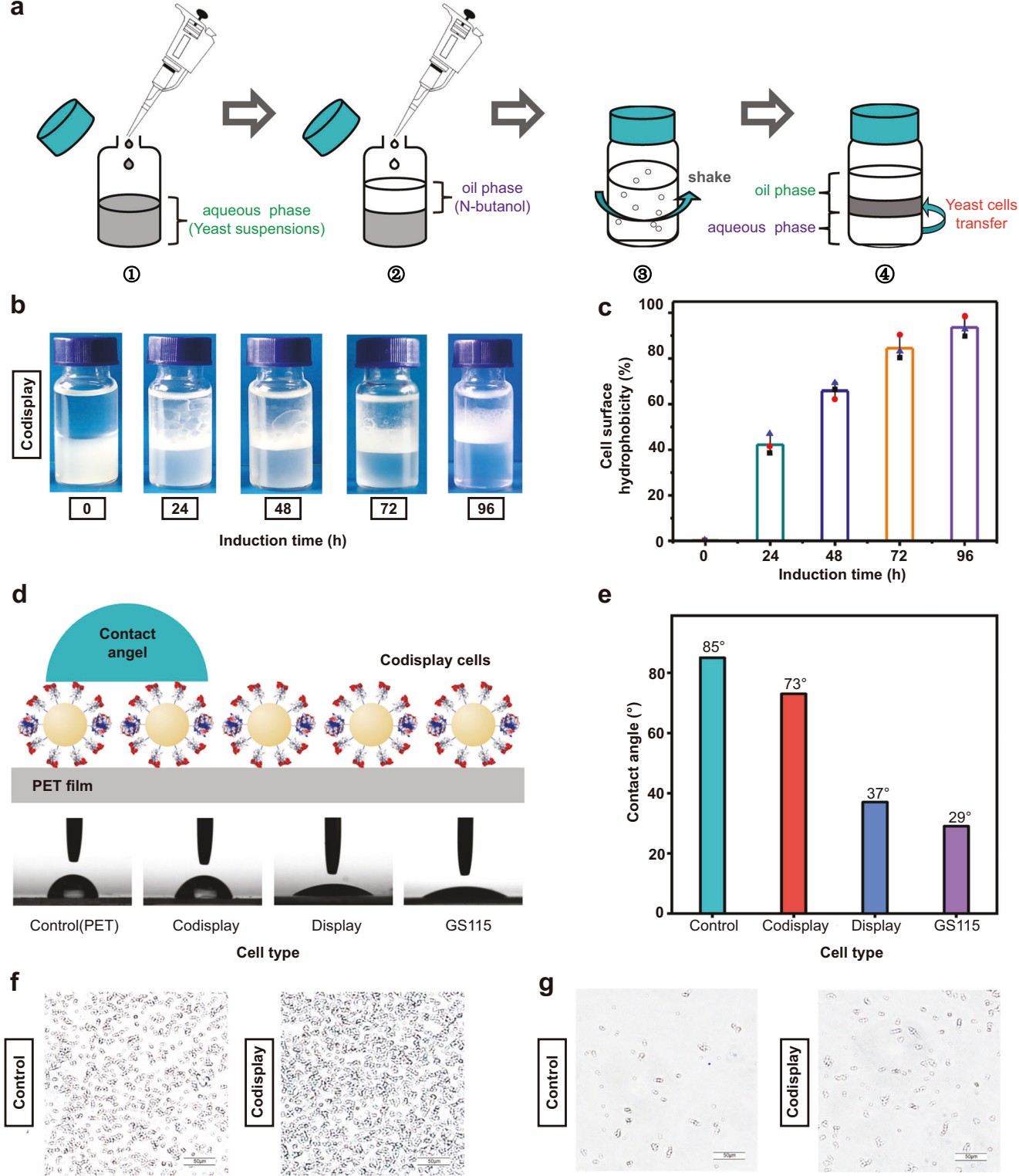

**Fig. 2 | Detection of the cell surface hydrophobicity. a** Schematic diagram of the MATH experiment. **b** Image the MATH experiment; **c** Measurement result of the cell surface hydrophobicity. *n* = 3 independent experiments. Data were presented as mean values ± SD. **d** Schematic diagram of the WCA experiment and image of the WCA experiment; **e** Measurement result of WCA. **f**, **g** The adsorption of the control sample-yeast cells displaying only PETase and the codisplay cells on the hcPET surface. The experiments are repeated three times independently, with similar results obtained. One representative is shown. Source data for panels (**c**, **e**) are provided as a Source Data file.

codisplay cells on the hcPET surface under different conditions. As shown in Fig. 2f, the codisplay cells covered almost all the surface areas of hcPET. However, under the same binding condition for the control sample-yeast cells that displayed only PETase, a sharply reduced

attachment to the PET surface was observed. The similar adsorption differences between the codisplay cells and cells displaying only PETase could also be observed when the same low concentration of yeast cells was used in the binding assays (Fig. 2g). We quantified the

yeast cell numbers adsorbed on the hcPET surface for all tested samples (Supplementary Fig. 10). It was quite clear that the binding numbers of the HFBI-displaying yeast cells on the hcPET surface were approximately twice that of the control group at each tested condition. The adsorption differences between the codisplay cells and cells displaying only PETase on the surface lcPET could also be observed, regardless of the concentration of yeast cells in the binding assays (Supplementary Fig. 11a, b). The number of HFBI-displaying yeast cells adsorbed on the lcPET surface was approximately twice that of the control group under each tested condition (Supplementary Fig. 11c, d). We believe that the similar surface hydrophobicity of hcPET and lcPET resulted in similar experimental results (Supplementary Fig. 12) by WCA. To further verify the role of HFBI in the codisplay system, we designed a codisplay system in which only hydrophobin was removed and the other parts remained unchanged. Supplementary Fig. 13a shows that both liker-GCW61 and PETase were successfully displayed on the surface of yeast cells. Then, we tested the enzymatic degradation performance against hcPET and lcPET, and we found that the codisplay system without hydrophobin exhibited a greatly reduced degradation ability toward both hcPET and lcPET compared with that of the normal codisplay system (Supplementary Fig. 13b, c). In addition, the adsorption capacity of this system on the PET substrate was reduced to approximately half of the original value (Supplementary Fig. 14). These results clearly revealed that the adhesive unit HFBI was particularly important for the codisplay system to exhibit a high degradation capacity to PET. Taken together, all these results confirmed that hydrophobin HFBI displayed on the yeast cell surface introduced the extra binding capacity to the yeast cell on the PET surface.

## The codisplay system exhibited extraordinary enzymatic activity against PET

We then explored the optimal conditions of the codisplay cells for hcPET hydrolysis and compared the results with those of purified PETase. Figure 3a and Supplementary Fig. 15a show that temperature exhibited an obvious influence on the turnover rates for both the codisplay cells and native PETase. The codisplay cells exhibited an enhanced enzymatic activity compared with that of native PETase at each tested temperature condition. The optimal reaction temperature was 40 °C, similar to native PETase, indicating that the codisplay cells remained in a heat-labile degradation system[14]. Since high temperature can promote the chain mobility of PET to accelerate its enzymatic degradation, a heat stabilizing unit could be introduced into the current surface display system to further improve its degradation efficiency in the future[18]. Figure 3b and Supplementary Fig. 15b shows the effect of pH on the turnover rates of the codisplay cells. The maximum turnover rate of the codisplay cells was achieved at pH 9, similar to native PETase. At other pH conditions, the codisplay cells exhibited much higher turnover rates than those of native PETase. For example, both yeast cells displaying only PETase (Supplementary Fig. 16) and native PETase were almost inactive at pH 10. However, the codisplay cells maintained more than 50% of their total enzyme activity, indicating that their alkaline tolerance was significantly improved. The surface-displayed HFBI may contribute to this result since HFBI is very stable at extreme pH conditions and can even resist protease degradation in vitro[44,62–64]. The increased tolerance of the codisplay cells to alkaline environments can facilitate the PET pretreatment process[65].

We then explored whether the cell number of codisplay cells influenced their turnover rate. Figure 3c shows that the turnover rate for the codisplay cells was at its maximum ($2146.5 \times 10^{-5} \, \text{sec}^{-1}$) and remained constant when the protein concentration was less than 1.3 nM. When the protein concentration exceeded 1.3 nM, the conversion rates decreased rapidly. This result revealed that there the displayed PETase on the yeast surface might be optimal, which corresponded to the maximum PET surface-reaching of the displayed

enzyme molecules. Additional displayed PETase molecules sterically hindered the PET surface and appeared inactive, leading to a decrease in the turnover rate of the codisplay cells. The native PETase and the display cells faced similar situations as the codisplay system, and the turnover rate reached maximum values ($6.5 \times 10^{-5}$ and $227.9 \times 10^{-5} \, \text{sec}^{-1}$) when the protein concentrations were 370 nM and 11.3 nM, respectively (Supplementary Fig. 16c and Supplementary Fig. 17). Finally, we found that the maximum turnover rate of the codisplay cells was strikingly 328.8-fold higher than that of native PETase (Fig. 3d).

Then, we performed both SEM and optical microscopy to observe the morphological changes in the codisplay cell-treated hcPET film. At the same time, we used native PETase-treated films as controls. As shown in Fig. 3e, there was almost no surface erosion on the PETase-treated hcPET film, suggesting that the enzymatic activity of native PETase against hcPET was rather low. In contrast, apparent cracks and erosion on the PET surface were observed by SEM when the yeast cells that codisplayed HFBI and PETase were applied to hcPET. Microscopic observation of cross-sections of the PET films was also performed to further evaluate the degradation results of our codisplay system. As shown in Fig. 3f-g and Supplementary Fig. 18, almost no change in the thickness of the PETase-treated hcPET film was observed. A small but obvious reduction in thickness (-3.2%) was observed when our codisplayed yeast cells degraded the hcPET film.

To further verify the functionality of the codisplay system, the enzymatic activity of the whole-cell biocatalyst toward a commercially available lcPET (PET-GF, crystallinity of 6.3%) was also measured. Considering that the degradation substrate was changed, we optimized the degradation conditions of both the codisplay cells and purified PETase for lcPET. As shown in Supplementary Fig. 19 and Supplementary Fig. 20, several reaction conditions, including temperature, pH, and cell number, caused obvious effects on the turnover rates of the codisplay cells and purified PETase. The optimal reaction temperature remained at 40 °C (Supplementary Fig. 19a), and the maximum turnover rate of the codisplay cells was achieved at pH 9, similar to native PETase (Supplementary Fig. 19b). These results were similar to those obtained by using hcPET as the degradation substrate. The turnover rate for the codisplay cells reached its maximum ($2909.3 \times 10^{-2} \, \text{sec}^{-1}$) when the normalized PETase concentration was 0.65 nM. In contrast, the turnover rate of purified lcPET reached a maximum value ($5.6 \times 10^{-2} \, \text{sec}^{-1}$) when the protein concentration was 181.4 nM (Supplementary Fig. 19c). Accordingly, the maximum turnover rate of the codisplay cells against lcPET was 519.5-fold higher than that of the purified PETase (Supplementary Fig. 19d).

Supplementary Fig. 21 shows the morphological changes in the codisplay cell-treated lcPET film, as observed by SEM and optical microscopy. Obvious erosion spots were observed on the surface of lcPET, which is consistent with the results of hcPET degradation, indicating that the results were statistically significant. In addition, it can be seen from the SEM results that the PET films degraded by the codisplay system were full of round pits with a size of ~5 microns, which was basically consistent with the size of yeast cells. These results revealed that yeast cells adsorbed on the PET surface and then exerted enzyme activity, as seen in the adsorption experiment (Supplementary Fig. 10 and Supplementary Fig. 11). The purified PETase contained small corrosion spots locally and was very small. This result indicated that the codisplay system could efficiently degrade lcPET and hcPET.

In addition, we tested the crystallinity of lcPET and hcPET before and after degradation at different temperatures using the codisplay system, as shown in Supplementary Fig. 22. For lcPET, it can be seen that the crystallinity increases with increasing conversion level, indicating that PETase degraded amorphous PET in the codisplay system. However, the crystallinity of the PET film treated with the system without enzyme at different temperatures did not change obviously compared with that of the control group, indicating that the

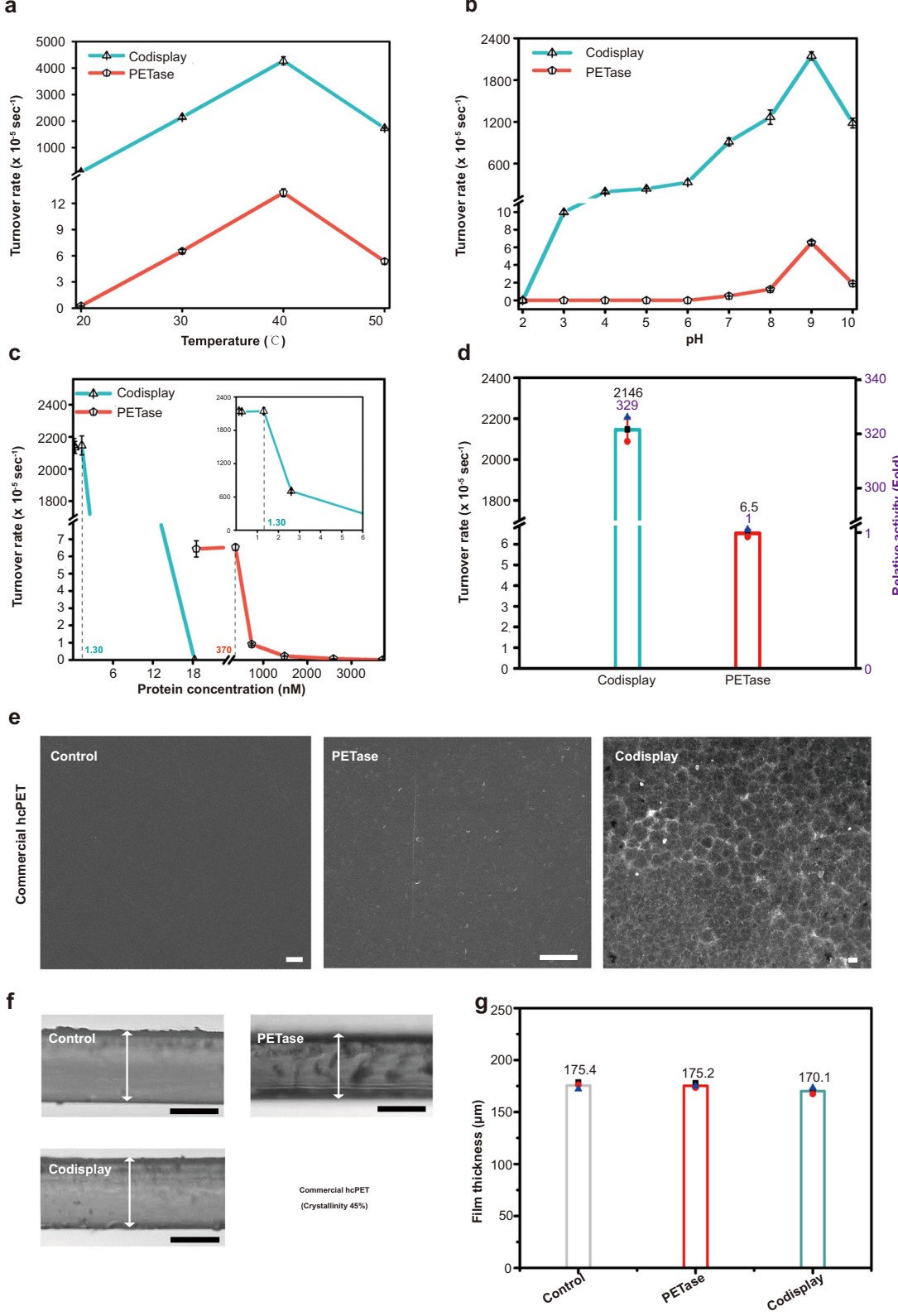

**Fig. 3 | Optimization of the degradation system and visualization of hcPET film degradation. a** Effect of temperature, **b** pH, and **c** protein concentration on PET hydrolysis. $n = 3$ independent experiments. Data were presented as mean values ± SD. **d** Comparison of the turnover rate of the codisplay cells and purified PETase under the optimal conditions using PET as a substrate. $n = 3$ independent experiments. Data were presented as mean values ± SD. **e** SEM image of commercial hcPET film before and after incubation with PETase and codisplay cells. (Scale bar:

5 µm). **f** Microscopic observation of a cross-section of a commercial hcPET film before and after incubation with PETase and codisplay cells. (Scale bar: 100 µm). **g** Measurement result of the cross-section of the commercial hcPET film. Data were presented as mean values ± SD. All the experiments are repeated three times independently, with similar results obtained. One representative is shown for (**e**) and (**f**). Source data for panels (**a**–**e**) and (**g**) are provided as a Source Data file.

temperature (range of 20–50 °C) exhibited little effect on the crystallinity of PET. The change in crystallinity observed in the experiment was mainly caused by enzyme digestion. The same results can also be observed in hcPET samples, except that the crystallinity decreases with increasing PET degradation, which suggested that the codisplay system exhibited no selectivity for PET hydrolysis. At the same time, the change in temperature did not affect the crystallinity of the hcPET film, indicating that the change in crystallinity of PET was also caused by PETase.

Finally, we compared the turnover rate based on the total products of MHET, TPA, and BHET for the purified PETase, yeast cells that displayed only PETase, and those with codisplay at the optimal condition using lcPET (Supplementary Fig. 23a) and hcPET (Supplementary Fig. 23b) as substrates. The conversion levels of hcPET and lcPET within 18 h at 30 °C were ~3.0 (depolymerization rate of 3.27 $mg_{products}$ $h^{-1}$ $mg_{enzyme}^{-1}$) and 55% (depolymerization rate of 178.15 $mg_{products}$ $h^{-1}$ $mg_{enzyme}^{-1}$), respectively (Supplementary Fig. 24).

## The codisplay whole-cell biocatalyst system is robust and efficient

It is expected that the codisplay cells will be used as whole-cell biocatalysts on an industrial scale; therefore, several properties related to industrial applications of the codisplay cells were evaluated, including thermal stability, reusability, chemical or solvent stability, storage conditions, and long-term enzymatic activity. Figure 4a shows that the relative turnover rate of the codisplay cells remained at 100% when the cells were placed at 30 °C for 7 days, indicating that the enzyme activity was almost unchanged during this incubation time. Free PETase lost 40% of its original hydrolytic activity after 1 day of incubation at 30 °C, and its enzyme activity was completely lost on the fifth day. These results revealed that the codisplay cells exhibited considerably higher thermostability[66,67]. This is one intrinsic advantage of the surface display system[68,69]. Sanna et al. reported that native HFBI did not degrade at 30 °C for 18 h[62]. Therefore, the excellent thermostability of HFBI might play a role in the thermostability of the codisplay cells. Then, the reusability of the codisplay cells was investigated.

As shown in Fig. 4b, the codisplay cells retained 85% of the original turnover rate after three rounds of use and 50% after seven cycles. There are two possible reasons for the decline in the turnover rate. The first reason is that cells are inevitably lost in the recycling process. The second reason is that the degradation products might occupy the catalytic pocket of the displayed PETase and inhibit its activities to some extent[70]. Next, we determined whether the presence of organic solvents or detergents would affect the turnover rate of codisplay cells due to the pretreatment process of PET, which often involves these chemicals. Figure 4c shows that the turnover rate toward hcPET retained 98, 82, and 72% when the codisplay cells were incubated in the tested solutions that contained 0.1% Triton X-100, 10% methanol, and 10% ethanol, respectively. Compared with the yeast cells that displayed PETase, the codisplay cells showed a higher turnover rate at each test chemical solution[34]. For example, the turnover rate of yeast cells that displayed PETase remained at just 50% in 0.1% Triton X-100 solution[34]. The chemical stability of the codisplay cells is beneficial to the PET pretreatment processes, such as cleaning[71]. It is well known that hydrophobin can maintain its functional activity under extreme conditions[42]. Hence, the chemical stability of hydrophobin HFBI may be related to the stability of the codisplay cells in the chemical solutions. Finally, the turnover rate of the whole-cell biocatalyst was evaluated before and after lyophilization, which is a dehydration process that facilitates the storage and transport of codisplay cells[72]. Supplementary Fig. 25 shows that the turnover rate of the codisplay cells toward PET remained nearly 100% after freeze-drying, suggesting that the whole-cell biocatalyst retained almost total enzymatic activity after the dehydration

process. Therefore, lyophilization is considered a storage option for the future use of whole-cell biocatalysts.

Next, we applied the codisplay cells to degrade commercial PET bottles. Figure 4d shows that the whole-cell biocatalyst can efficiently degrade different highly crystallized PET bottles. According to a previous method[13], we calculated the conversion level of the codisplay cells against the highly crystallized PET bottles used in our study. As shown in Table 1, the conversion levels were 3.1, 3.1, and 2.8% for Nestle, Coca-Cola, and Pepsi PET bottles, respectively. In Yoshida's paper, the conversion level of native PETase against hcPET was ~0.0097%[14]. Compared to native PETase, the codisplay cells exhibited much higher degradation efficiency in the highly crystallized PET bottles. All were consistent with the conversion levels for hcPET and lcPET. The conversion level of the codisplay system was 3 and 55% toward hcPET and lcPET, respectively, which was consistent with previous microscopic observations. We also summarized and calculated the conversion levels of several reported PETase mutants using the same procedure in the Methods section (Table 1). Clearly, the codisplay system showed the best conversion level under 18 h for both high- and low-crystallinity PET. Finally, to evaluate the long-term functionality of the designed whole-cell biocatalyst, we used the codisplay system to degrade hcPET for 10 days. As shown in Fig. 4e, f, the total products and relative conversion levels of hcPET in both the codisplay system and the display system increased with time and reached a maximum around the ninth day, which was 10.9 and 1.2%, respectively. However, the degradation activity of wild-type PETase for hcPET was very low, and the conversion level was only 0.003%. These results showed that by establishing the codisplay system, the stability of PETase increases.

Although the conversion level of codisplay cells against the commercial bottles was the highest ever reported for any degradation system using PETase to our knowledge, there is still much room for improvement. For example, we can screen and test more adhesive units to further improve the binding of the yeast cell onto the PET surface. By doing so, we hope that the codisplay system can be applied in the large-scale biological recycling of PET in the future.

## Molecular insights into the two-step degradation of PET by the codisplay system

To explore how membrane-anchored HFBI/PETase engaged PET chains and how these interactions facilitated the subsequent cleavage of PET, all-atom MD simulations were performed. MD simulation has been adopted by several researchers to investigate the interactions between wild-type or mutant PETase and their substrate PET in soluble states[11,73–77]. In the preequilibrated system (0 ns), randomly distributed 4PET did not interact with the active center of PETase or the hydrophobic region of HFBI (Fig. 5a, Supplementary Fig. 26, and Supplementary Movies 1, 2). Over time, 4PET gradually aggregated near the hydrophobic amino acids of HFBI at ~40–50 ns. At ~60–70 ns, 4PET completely attached to the hydrophobic patch of HFBI, which occurred mainly through the interaction of hydrophobic residues, and remained mostly bound for the rest of the trajectory. During this period, 4PET began to gather near the active center of PETase; eventually, a number of 4PET molecules were gathered at ~100 ns. Calculation of the distances between 4PET and the other two proteins agreed with the possible role of HFBI in the local enrichment of PETs (Supplementary Fig. 27). From the simulated trajectories, we postulated that our codisplay system completed PET degradation through two-step processes involving adsorption and hydrolysis, wherein the hydrophobin HFBI favorably binds to the PET chains and grabs the chains firmly (Fig. 5b); then, PETase interacts with PET and performs a hydrolysis function (Fig. 5c).

Next, to delineate the interaction between PETase-linker-GCW51 and hcPET and to elucidate whether PETase-linker-GCW51 follows a conformational selection or induced-fit mechanism, we

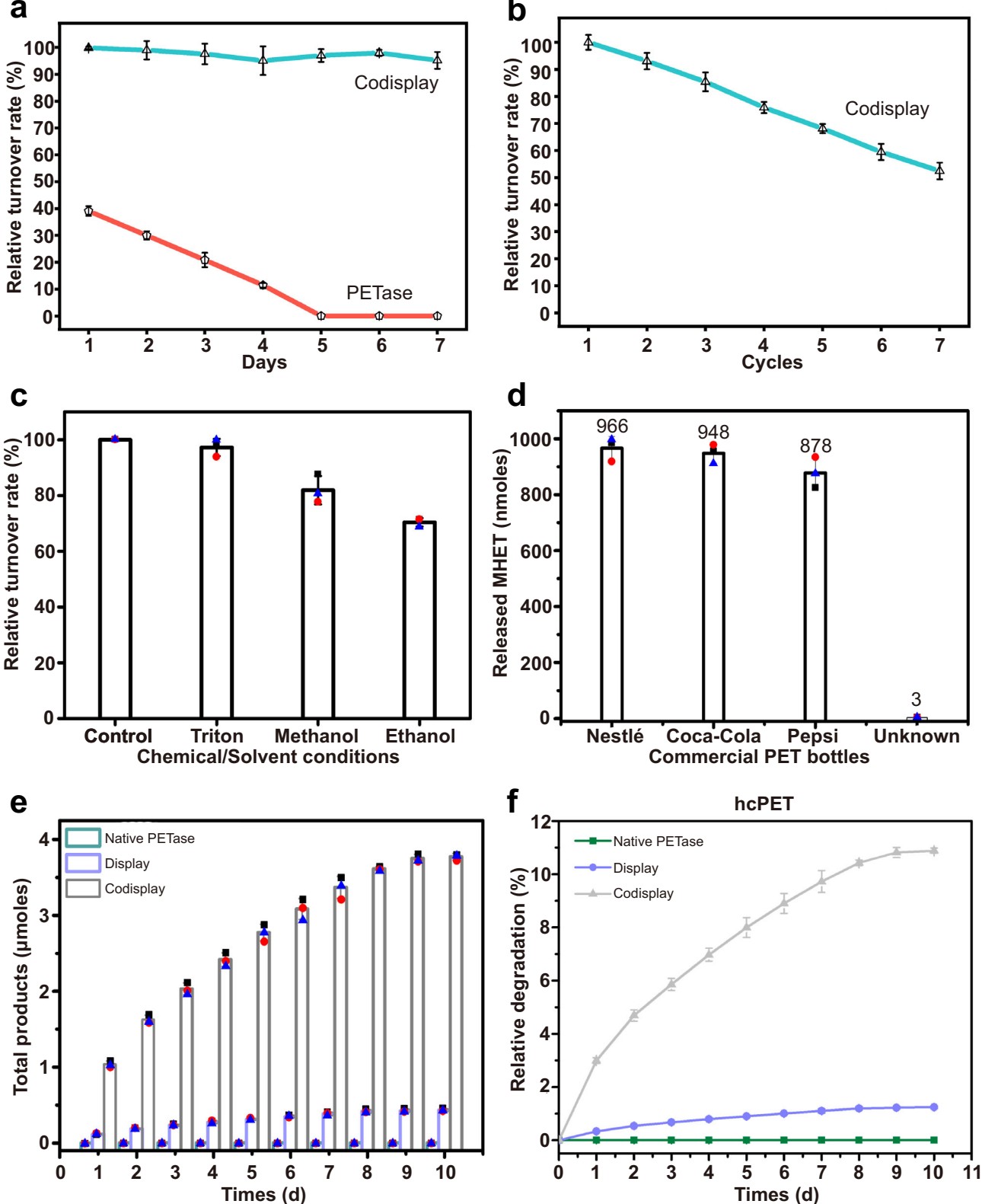

**Fig. 4 | The stability and functionality of the codisplay system. a** The thermostability of GS115/PETase-HFBI and purified PETase. **b** Effect of recycling times by *P. pastoris* GS115/PETase-HFBI. **c** Effect of chemical/solvent conditions on *P. pastoris* GS115/PETase-HFBI and GS115/PETase-GCW51. **d** Degradation of commercial PET bottles with the whole-cell biocatalyst. The first to third columns of data were measured by using the codisplay whole-cell biocatalyst, and the fourth column of data was obtained from ref. 14. The total products (**e**) and relative conversion levels (**f**) of the hcPET degraded by purified PETase, *P. pastoris* GS115/PETase-GCW51 and GS115/PETase-HFBI for a long reaction time. All the experiments are repeated three times. Data were presented as mean values ± SD. Source data are provided as a Source Data file.

**Table 1 | Comparison of the degradation abilities of PETase and its mutants toward PET films**

| Degradation system | PET film (crystallinity) | Reaction conditions | Total products (µmoles) | Conversion level (%)[a] | Normalized Conversion level (18 h) | References |
|---|---|---|---|---|---|---|
| Codisplay | 45% | 30 °C 18 h | 0.92 | 3.0 | 3.0% | Our work |
| | Nestle | 30 °C 18 h | 0.97 | 3.1 | 3.1% | Our work |
| | Coca-Cola | 30 °C 18 h | 0.95 | 3.1 | 3.1% | Our work |
| | Pepsi | 30 °C 18 h | 0.88 | 2.8 | 2.8% | Our work |
| S121E/D186H/R280A | 41.79% | 30 °C 72 h | 0.018 | 0.041 | 0.015% | [18] |
| | 41.79% | 30 °C 72 h | 0.037 | 0.085 | 0.04% | [18] |
| S121E/D186H/S242T/N246D | 41.79% | 37 °C 20 day | 0.24 | 0.55 | 0.021% | [20] |
| IsPETase | 30% | 30 °C 18 h | 0.0045 | 0.0097 | 0.0097% | [14] |
| DuraPETase | 30% | 37 °C 10 day | 1.55 | 15 ± 1 | 1.38% | [17] |
| S238F/W159H | 14.8 ± 0.2% | 30 °C 96 h | 0.62 | 1.43 | 0.27% | [11] |
| Codisplay | Gf-PET 6.3% | 30 °C 18 h | 25.05 | 55 | 55% | Our work |
| TS-PETase | Gf-PET 6.3% | 30 °C 6 day | 4.2 | 9.61 | 1.20% | [21] |
| R280A | Unknown | 30 °C 18 h | 0.0055 | 0.013 | 0.013% | [15] |

[a]The calculation method is from Table S2 in Fusako Kawai et al. 2019.

performed induced fit molecular docking (IFD) analysis of PETase-linker-GCW51, in which a PET-tetramer represented the polymer substrate (4PET)[73]. The five highest-scoring docking poses were subjected to analysis with respect to the spatial arrangement of the residues of the catalytic triad and the conformation of the reactive part of the oligomer substrate[73]. We found the best-predicted docking pose while taking the productivity of the catalytic triad into account, as shown in Fig. 5c. The reacting carbonyl carbon of the substrate was bound with the scissile ethylene glycol moiety in the canonical gauche conformation ($\Psi_{gauche}$) in a twisted chain conformation, which was consistent with how that wild-type PETase binds to 4PET[73].

In this study, we developed a codisplay system to mimic the natural two-step gradation process of PET by a whole-cell biocatalyst. Among the codisplay systems, the hydrophobin HFBI displayed on the yeast cell surface was proposed to function as an adhesive unit to enhance the attachment of codisplay cells to the PET surface, which would achieve the aim of efficient degradation of hcPET by the whole-cell biocatalyst. Our data showed that the displayed HFBI causes the codisplay cells to exhibit a hydrophobic surface property. A twofold increase in the binding numbers of the hydrophobin-producing yeast to hydrophobic PET was observed under the optimal catalytic conditions. Hydrophobic interactions between yeast cells and their substrate PET may play a major role in attachment. An enzymatic assay confirmed that the enzymatic activity of the codisplay cells was ~328.8 times higher than that of the purified PETase. Our results not only provide an efficient strategy for efficiently biodegrading hcPET but also demonstrate the plasticity of the surface display system. By introducing different functional modules into the surface display system, its performance can be greatly improved. In the future, more functional units, such as MHETase, that can hydrolyze MHET to PET reagents can be introduced into the whole-cell biocatalyst system to further enhance its performance.

## Methods
### Strains and culture conditions
The microbial strains and plasmids used in this study are listed in Supplementary Table 2. *Pichia pastoris* (*P. pastoris*) strain GS115 and *Escherichia coli* (*E. coli*) DH5α were stored in our laboratory. *E. coli* DH5α cells were used in pPICZαA plasmid construction and were incubated at 37 °C in Luria broth (LB) low salt medium (1% w/v tryptone, 0.5% w/v yeast extract, and 0.5% w/v NaCl) supplemented with

100 µg/mL zeocin and were also used in pPIC9 plasmid construction and were incubated at 37 °C in LB medium (1% w/v tryptone, 0.5% w/v yeast extract, and 1% w/v NaCl) supplemented with 100 µg/mL ampicillin. *P. pastoris* yeast strains were cultured at 30 °C in the following media: YPD (1% w/v yeast extract, 2% w/v peptone, and 2% w/v glucose) for subcultivation, BMGY (1% w/v yeast extract, 2% w/v peptone, 100 mM potassium phosphate pH 6.0, 1.34% w/v yeast nitrogen base (YNB), and 1% v/v glycerol) for cell growth, and BMMY (same as BMGY but 1% v/v glycerol was substituted for 1% v/v methanol) for recombinant protein production.

### Amplification of target genes
The codon-optimized PETase gene sequence (Supplementary Sequence 1) and the *T. reesei* HFBI gene sequence (GenBank KU173825) were synthesized by the BGI Group (China). All primers used for plasmid construction, which were synthesized by GENEWIZ (China), are listed in Supplementary Table 3. The *gcw51* gene (NCBI accession no. XM_002493737.1) was amplified by polymerase chain reaction (PCR) from the genomic DNA of *P. pastoris* GS115 using primers 51-F/51-R, which contained an overlap area at the 5′-terminus and an *Eco*R I site at the 3′-terminus. The *petase* gene was amplified using the PCR method with primers P-F/P-R, containing an *Xho* I site, a flag tag at the 5′-terminus, and an overlap area at the 3′-terminus. The *gcw61* gene (NCBI accession no. XM_002494287.1) was amplified by PCR from the genomic DNA of *P. pastoris* GS115 using primers 61-F/61-R, containing a *Not* I site at the 3′-terminus and an overlap area at the 5′-terminus. The *hfb I* gene was amplified using the same method with primers hf-F/hf-R, which contained an *Eco*R I site at the 5′-terminus and an overlap area at the 3′-terminus. All PCR products were gel-purified, and then the fusion fragments PETase-GCW51 and HFBI-GCW61 were successfully built through overlap PCR. The petase-linker-gcw51 and petase-linker genes were cloned from the already constructed GS115/PETase-HFBI using the restriction sites *Nde* I and *Xho* I by the primers p51-F/p51-R and PL-F/PL-R, respectively. Then, the gene was cloned into the pET-21a(+) vector using the same restriction sites.

### Vector construction and yeast transformation
The fusion fragment petase-gcw51 and the vector pPIC9 were digested with *Xho* I and *Eco*R I at 37 °C for 1 h, respectively. Then, the gene and vector were ligated in the ligation system at 16 °C overnight to obtain the recombinant plasmid pPIC9/petase-gcw51. The fusion fragments hfbI-gcw61, linker-gcw61 and the vector pPICZαA were digested with *Eco*R I and *Not* I at 37 °C for 1 h, respectively. Then, the two genes and

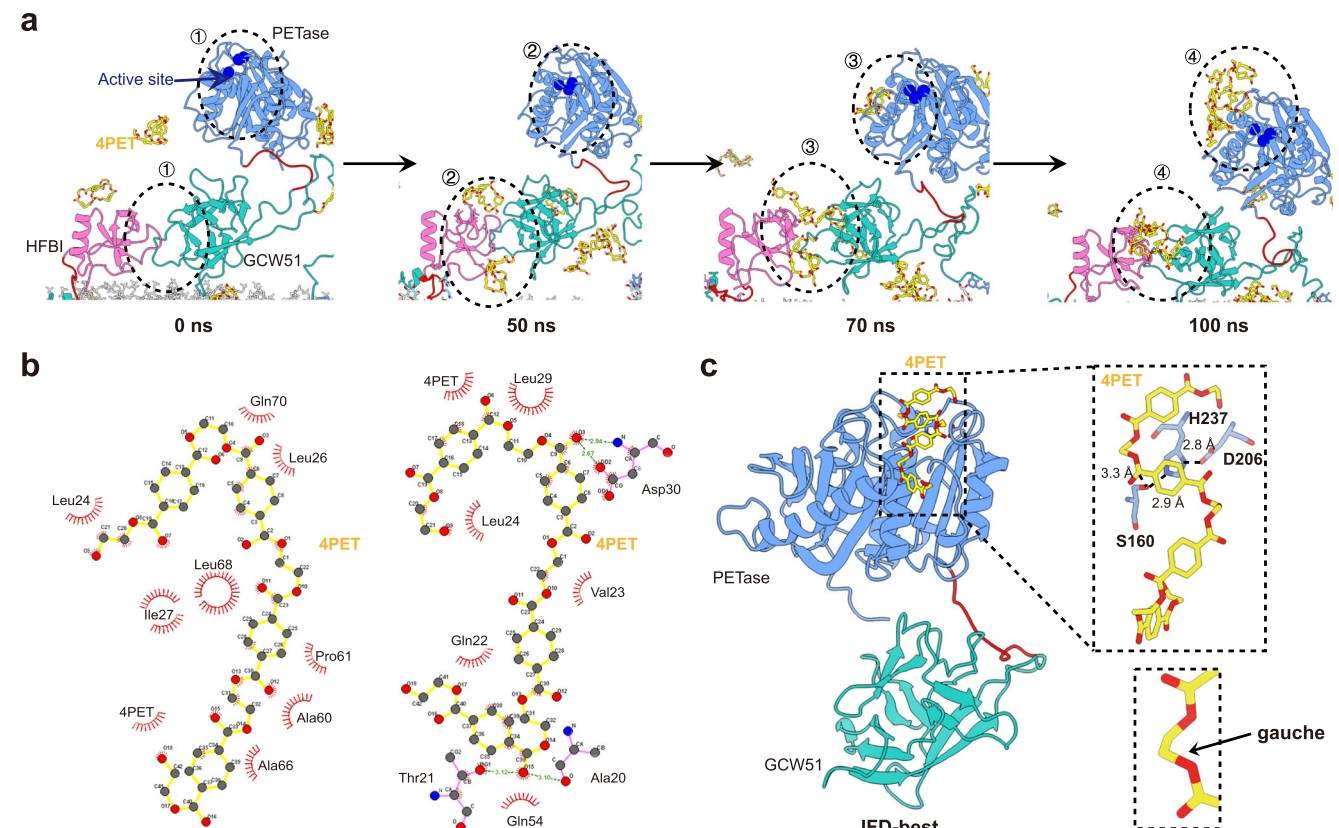

**Fig. 5 | Proposed two-step mechanism of PET degradation by the codisplay system. a** MD simulation of the dynamic process of PET adsorption by the codisplay system. **b** Hydrophobic interaction between PET substrate and HFBI. **c** IFD analysis of the conformational fitting for the codisplay system during hydrolysis. Each domain is assigned a unique color. The active sites are marked with blue spheres. 4PET are marked with sticks.

vector were ligated in the ligation system at 16 °C overnight to obtain the recombinant plasmids pPICZαA/hfbI-gcw61 and pPICZαA/linker-gcw61.

To display PETase and HFBI on the cell surface, the recombinant plasmids pPIC9/petase-gcw51 were then linearized with *Stu* I, and pPICZαA/hfbI-gcw61 and pPICZαA/linker-gcw61 were then linearized with *Sac* I. Then, the linearized products were subsequently transformed into the host strain GS115 by electroporation.

Finally, the genome of the positive transformants selected from YPD plates containing 100 μg/mL Zeocin as the selective marker was used as a template, and PCR was carried out using the target gene primers and AOX primers to screen three kinds of true positive transformants, GS115/PETase-HFBI, GS115/PETase, and GS115/PETase-ΔHFBI.

**Western blot and immunofluorescence microscopy analysis**
The purified Flag-tagged PETase protein was used to prepare protein standard curves. A nanodrop was used to accurately measure the PETase concentration. Proteins (10, 50, 200, and 300 ng) were accurately obtained for SDS-PAGE and transferred to PVDF Immobilon-P transfer membranes (0.45 μM pore size, Sigma-Aldrich) under 100 V for 1 h for use in immunoblot analyses. The antibodies used for Western blotting were primary rabbit anti-FLAG antibody (ABclonal, catalog# AE092, 1:2000), primary mouse HFBI antibody (storage in our laboratory, 1:1000), primary mouse anti-His antibody (SUNGENE BIOTECH, catalog# LK8001, 1:1000), goat anti-rabbit IgG antibody (SUNGENE BIOTECH, catalog# LK2001, 1:2000), and goat anti-mouse antibody (SUNGENE BIOTECH, catalog# LK2003, 1:2000). All serial Western blot data were representative of at least three biological experiments. The chemiluminescent substrate (Thermo Fisher Scientific) was added for imaging detection (GelDoc XR+, Bio-Rad, USA).

Then, ImageJ was used to quantify the grayscale of the lane and draw the standard curve.

To determine the soluble and nonsoluble ratio of PETase expressed in the yeast strain, we performed Western blot analysis on whole cells, cell walls, and protoplasts. Before the samples were prepared for SDS-PAGE, the $OD_{600}$ value of the yeast solution was measured, and the cell counts were all calculated as the same as $2 \times 10^8$.

For the whole-cell proteins, the cells were disrupted in 50 μL buffer A (20 mM Tris/HCl, pH 7.5, 200 mM NaCl) by a bead beater (SI-D258, Scientific Industrial) at 4 °C. Then, 5× SDS-PAGE loading buffer (E153-01, GenStar, China) was added to a final concentration of 1×. After being boiled at 100 °C for 10 min, the samples were centrifuged at 13,000×*g* for 15 min, and the supernatants were collected and resolved by SDS-PAGE.

For the cell wall proteins, the cells were disrupted in 50 μL buffer A (20 mM Tris/HCl, pH 7.5, 200 mM NaCl) by a bead beater (SI-D258, Scientific Industrial) at 4 °C. Then, 150 μL of 1% (v/v) Triton X-100 was added to extract cell wall proteins at 4 °C for 30 min[78,79]. 5× SDS-PAGE loading buffer (E153-01, GenStar, China) was added to a final concentration of 1×. After being boiled at 100 °C for 10 min, the samples were centrifuged at 13,000×*g* for 15 min, and the supernatants were collected and resolved by SDS-PAGE.

For protoplast proteins, the cells were resuspended in 600 μL sorbitol buffer, and 25 U lyticase (Tiangen) was added and incubated at 4 °C for 30 min. Samples were centrifuged at 1500×*g* for 10 min, and the cell pellets were collected[80]. Subsequently, the cell pellets were disrupted in 50 μL buffer A (20 mM Tris/HCl, pH 7.5, 200 mM NaCl) by a bead beater (SI-D258, Scientific Industrial) at 4 °C. Then, 5× SDS-PAGE loading buffer (E153-01, GenStar, China) was added to a final concentration of 1×. After being boiled at 100 °C for 10 min, the samples were centrifuged at 13,000×*g* for 15 min, and the supernatants were

collected and resolved by SDS-PAGE. Then, the grayscale of the lane was used to quantify and calculate the proportion of protein in each cell component.

To quantify the relationship between the number of cells and the concentration of PETase in the codisplay system and display system, we used $2 \times 10^7$ yeast cells after 48 h of induction for SDS-PAGE electrophoresis and quantitative analysis by Western blotting. According to the standard curve, for the display system, the PETase expression amount of $2 \times 10^7$ cells was equivalent to 151.2 ng. For the Codisplay system, the PETase expression amount of $2 \times 10^7$ cells was equivalent to 35.2 ng. PET degradation experiments in this study were carried out by taking corresponding multiples of $2 \times 10^7$ yeast cells, and the concentration of PETase was calculated according to the proportion.

The antibodies used for immunofluorescence microscopy analysis were FITC goat anti-mouse IgG (H + L) (Abclonal, catalog# AS001, 1:200) and rhodamine (TRITC) goat anti-rabbit IgG (H + L) (Abclonal, catalog# AS040, 1:200). To confirm the expression and localization of PETase-GCW51 and HFBI-GCW61 fusion proteins on the yeast cell surface, the yeast cells were visualized on a confocal microscope (UltraView Vox, PerkinElmer, USA).

## Protein expression and purification

PETase-linker and PETase-linker-GCW51 were expressed in *E. coli* BL21(DE3) in Luria broth (LB) at 16 °C for 16 to 18 h with 0.5 mM iso-propyl β-D-1-thiogalactopyranoside (IPTG). The purification procedures are described as follows:

Bacteria expressing proteins were harvested and resuspended in a lysis buffer containing 20 mM Tris·HCl pH 7.5, 0.3 M NaCl, and 10% glycerol and lysed by high-pressure homogenization. After centrifugation ($16,000 \times g$, 30 min at 4 °C), the supernatant was loaded onto a nickel nitrilotriacetic acid (Ni-NTA) column (GE Healthcare). The column was washed using lysis buffer supplemented with 30 mM imidazole and eluted using lysis buffer supplemented with 300 mM imidazole. Finally, the protein was purified by gel-filtration chromatography (Superdex 75 10/300 GL, GE Healthcare) using a buffer containing 20 mM Tris·HCl pH 7.5, 0.3 M NaCl.

## Crystallization, data collection, and structure determination

All crystals were grown by the microbatch-underoil method unless otherwise specified. PETase was crystallized at 16 °C by mixing 1 μL of protein (15 mg/mL) with 1 μL of crystallization buffer containing 0.2 M calcium chloride dihydrate, 0.1 M HEPES sodium pH 7.5, 28% v/v polyethylene glycol 400. The crystal of PETase-linker was grown at 16 °C from a mixture of 1 μL protein (15 mg/mL) and 1 μL crystallization buffer containing 10% w/v PEG 8000, 100 mM MES/sodium hydroxide (pH 6.0) and 200 mM zinc acetate. The crystals were cryoprotected by Parabar 10312 (previously known as paratone oil). X-ray diffraction data were collected on beamline BL19U1 at the Shanghai Synchrotron Radiation Facility at 100 K and at a wavelength of 0.97861 Å. Data integration and scaling were performed using HKL3000. The wild-type PETase and PETase-linker structures were all solved by molecular replacement using the structure of *Is*PETase (PDB ID: 5XJH)[15] as a search model through the PHASER program from the CCP4 package. Model building and refinement were performed using PHENIX (version 1.14) and COOT (version 0.8.9).

## Enzyme activity assay for PET

To evaluate the hydrolytic activity of PETase and the codisplayed recombinant *P. pastoris*, the hcPET film (Good Fellow, crystallinity, 45%, thickness, 0.175 mm, diameter, 6 mm) and the lcPET film (Goodfellow Cambridge, PET-GF, the crystallinity of 6.3%, 0.25 mm thick, diameter, 6 mm) were used as the substrates for degradation assays with the purified PETase enzyme and the displayed system. Before the reaction, the PET film was soaked separately in 0.5% Triton X-100, 10 mM Na$_2$CO$_3$,

and distilled water (each was performed at 50 °C for 30 min at 550 rpm), and then air-dried for the reaction. Subsequently, the PET film was placed into a tube with 300 μL buffer containing 50 mM glycine-NaOH (pH 9.0) for 18 h at 30 °C with 370 nM purified enzyme and corresponding yeast cells. To optimize the induction time, purified PETase- and GS115/PETase-HFBI-displayed yeast cells were induced for 0, 24, 48, 72, and 96 h for the enzyme activity assay. After removing the enzyme-treated PET film from the reaction mixture, the enzyme reaction was terminated by heating at 85 °C for 10 min. The reaction mixture samples were then centrifuged at $12,000 \times g$ for 5 min. The supernatant of each sample was further analyzed by high-performance liquid chromatography (HPLC) to quantify the PET monomers released from PET depolymerization. To compare the PET hydrolytic activity of the displayed system with WT PETase across a range of pH values (2.0–10.0) at 20 and 50 °C, a similar experimental setup was used.

Then, the enzymatic activity of the codisplayed PETase was determined by calculating the turnover rate of the codisplayed PETase in each experiment. The turnover rate was calculated by normalizing the amount (moles) of the total products (MHET, BHET, TPA) generated in each experiment to the amount (moles) of PETase present in each experiment, and then the resultant ratio was further divided by the enzymatic degradation time ($18 \times 3600$ s). All the following enzymatic activities of codisplayed or native PETase were determined in the same way.

For the long-term enzyme activity reaction, the PET film was placed into a tube with 300 μL buffer containing 50 mM glycine-NaOH (pH 9.0) at 30 °C with 370 nM enzyme. Then, samples were taken every 24 h for 10 consecutive days, and the amount of the three products was detected by HPLC to calculate the conversion level. A similar experimental setup was used for the displayed system.

The specific activity (SA)[9] of the enzyme during the PET depolymerization reaction, in mg of equivalent products generated per h per mg of enzyme ($\text{mg}_{products}\ h^{-1}\ \text{mg}_{enzyme}^{-1}$) for 18 h reactions (1), and in mg of equivalent total products generated per d per mg of enzyme ($\text{mg}_{products}\ d^{-1}\ \text{mg}_{enzyme}^{-1}$) for prolonged reactions (2), was determined by monitoring the liberation of the total products of terephthalic acid (TPA), mono(ethylene terephthalate) (MHET) and bis(2-hydroxyethyl) terephthalate (BHET). TPA, MHET, and BHET were measured according to standard curves drawn below, as prepared from commercial TPA and BHET.

$$SA_1 = \frac{\triangle m_{PET}}{18 \times m_{PETase}} \tag{1}$$

$$SA_2 = \frac{\triangle m_{PET}}{10 \times m_{PETase}} \tag{2}$$

## HPLC analysis of the degradation product of PET

HPLC was performed on a Waters e2695 equipped with a HyPURITY C18 (Thermo Fisher Scientific, No 22105-254630) column (4.6 × 250 mm). The mobile phase was methanol/18 mM phosphate buffer (pH 2.5) at a flow rate of 0.5 mL min$^{-1}$, and the effluent was monitored at a wavelength of 240 nm. The typical elution conditions were as follows: 0 to 30 min, 25% (v/v) methanol, and 30 to 50 min, 25–100% methanol linear gradient. The total peak areas of MHET, TPA, and BHET were used to calculate the amount of products in each PET hydrolysis reaction.

Standard MHET was obtained from the complete hydrolysis of BHET, which was purchased from Sigma. The detailed process was as follows. Four millimolar BHET was incubated with 50 nM PETase in 40 mM Na$_2$HPO$_4$·HCl (pH 7.0), 80 mM NaCl, and 20% (v/v) DMSO at 30 °C. After the complete hydrolysis of BHET to MHET was confirmed by HPLC, the protein was removed from the reaction mixture with

Amicon Ultra 10 kDa (Merck Millipore), resulting in a 4 mM MHET solution.

Then, MHET, TPA, and BHET of different concentrations were used to run HPLC, and the standard curves of the three were drawn according to the relationship between peak area and sample loading.

### Cell surface hydrophobicity assay
The hydrophobicity of yeast cell surfaces was determined by using microbial adhesion to hydrocarbons (MATH), which was modified from a previous protocol[55]. The induction time was 0, 24, 48, 72, and 96 h, and the yeast cell suspension was washed twice with PUM buffer (containing 150 mM phosphate, potassium, and magnesium, pH 7.1). Then, the absorbance of the cells in the PUM buffer was adjusted to $OD_{600} = 4$ and defined as $A_1$. The cell suspension (0.75 mL) was added to a 2 mL glass bottle, then 0.75 mL of *n*-butanol was added, and the cap was tightly closed. After 30 s of an eddy, the mixture was left for 3 min to complete the two-phase separation. The absorbance of the water phase at 600 nm was determined as $A_2$. The hydrophobicity is given as a percentage (%) = $(A_1 - A_2)/A_1$.

### Contact angle measurements
Yeast cells were harvested by centrifugation, washed once with Milli-Q water, and finally resuspended in pure ethanol at a concentration of $10^8$ cells mL$^{-1}$. A small volume of the cell suspension (250 μL) was spread over the PET film. After drying the first layer of cells, two more layers were added, completely covering the PET film. Contact angles were measured on agar plates covered with cells. The measurements were carried out in a standard contact angle apparatus (KSV Instruments Ltd, CAM 200). The contact angles were determined automatically with the aid of an image analysis system.

### Adsorption assay of cells on PET
Each PET film (diameter, 6 mm) was placed in $2 \times 10^6$, $4 \times 10^6$, $2 \times 10^7$, $4 \times 10^7$, and $1.2 \times 10^8$ mL$^{-1}$ cell suspensions and shaken at 30 °C, and the PET film was removed after 18 h for microscopic examination (BX51, Olympus, Japan).

### Optimization of the codisplay system
To optimize the pH, the reactions were conducted in 50 mM NaH$_2$PO$_4$-NaOH (pH 4.0–5.0), 50 mM Na$_2$HPO$_4$-HCl (pH 6.0–8.0), or 50 mM glycine-NaOH (pH 9.0–10). To optimize the temperature, the reactions were conducted at 20, 30, 40, and 50 °C.

### Standardizing the cell number to protein concentration
To quantify the displayed PETase concentration, we used Western blotting for quantitative analysis. First, we used PETase for Western blot grayscale analysis and plotted a standard curve, as shown in Supplementary Fig. 6a, b. Subsequently, we used $2 \times 10^7$ yeast cells after 48 h of induction for SDS–PAGE electrophoresis and quantitative analysis by Western blot, as shown in Supplementary Fig. 6c. According to the standard curve, for the display system, the PETase expression amount of $2 \times 10^7$ cells was equivalent to 151.2 ng. For the Codisplay system, the PETase expression amount of $2 \times 10^7$ cells was equivalent to 35.2 ng. PET degradation experiments in the manuscript were carried out by taking corresponding multiples of $2 \times 10^7$ yeast cells, and the concentration of PETase was calculated according to the proportion.

### Optimization of protein concentration for the PET hydrolysis reaction
The PETase protein concentrations were 0.5, 1, 5, 10, 20, 40, 70, and 100 μg mL$^{-1}$. The MHET quality of the PETase reaction with PET at 10 μg mL$^{-1}$ was defined as 100%, and the relative enzyme activity was calculated. The GS115/PETase-HFBI cell concentration was standardized to $2 \times 10^6$, $4 \times 10^6$, $2 \times 10^7$, $4 \times 10^7$, $1.2 \times 10^8$, $2 \times 10^8$, $2.8 \times 10^8$, and

$4 \times 10^8$ mL$^{-1}$. All reactions were performed in 50 mM glycine-NaOH (pH 9.0) at 30 °C for 18 h.

### Scanning electron microscopy (SEM)
Each PET film (diameter, 6 mm) was examined by SEM, both before and after degradation treatment, under the same conditions. PET samples were rinsed with 1% SDS, then rinsed with Milli-Q water and ethanol, sputter-coated with Au, and mounted on aluminum stubs using carbon tape. SEM imaging was performed using an FEI Quanta 400 FEG instrument under a low vacuum operating with a gaseous solid-state detector. Imaging was performed with a beam-accelerating voltage of 5 keV.

### Observing the thickness change of PET film before and after degradation
To measure the change in PET film thickness, the hydrolyzed PET film after each incubation was transferred into a 1.5 mL tube with buffer (1 mL, 100 mm Tris-HCl, pH 6.8) containing 5% w/v SDS to remove the adsorbed protein and cells and was washed with Milli-Q water. The washed films were cut in half to measure the cross-section and set vertically on the stage of an inverted microscope equipped with a V10 ocular and a V10 objective. The cross-section was observed in the bright field. The thickness was calculated from the number of pixels in the image based on the scale bar.

### Reusability and stability assays of the whole-cell biocatalyst
For the thermostability of yeast, the residual yeast activity was measured after incubation at 30 °C for 1–7 days in 50 mM glycine-NaOH buffer (pH 9.0). The yeast was recycled after centrifugation at $3500 \times g$ for 5 min. For the chemical/solvent condition's stability of yeast, the residual yeast activity was measured after incubation in 50 mM glycine-NaOH (pH 9.0) supplemented with methanol (final concentration, 10% v/v), ethanol (final concentration, 10% v/v), DMSO (final concentration, 10% v/v), Tween (final concentration, 0.1% v/v), and Triton X-100 (final concentration, 0.1% v/v) for 18 h at 30 °C. Then, the PET reaction was started after washing 3 times with 50 mM glycine-NaOH (pH 9.0) at $3500 \times g$ for 5 min.

### Effect of freeze-drying on PET film hydrolysis
For the freeze-dried yeast, the freeze-dried yeast activity was measured after incubation at 4 °C overnight in 50 mM glycine-NaOH buffer (pH 9.0). The $OD_{600}$ value was measured to ensure that the cell count was the same as that before freeze-drying.

### Enzyme assays for commercial PET bottles
GS115/PETase-HFBI cells (induced for 48 h, total cell number is $6.2 \times 10^9$) were incubated with 6 mm diameter hcPET cut out from different commercial PET bottles (brands: Coca-Cola, Nestlé, and Pepsi) in 50 mM glycine-NaOH buffer (pH 9.0) for 18 h at 30 °C.

### Growth curve measurement
The transformant GS115/PETase-HFBI was inoculated into BMGY medium at 30 °C to an $OD_{600}$ of 2 to 6. The culture was centrifuged at $3500 \times g$ for 5 min and resuspended in BMMY medium to an $OD_{600}$ of 1. To induce the fusion protein for expression, cells were incubated at 30 °C with 100% methanol every 24 h to a final concentration of 1%. *Pichia pastoris* GS115 was used as a negative control. The $OD_{600}$ was measured every 24 h from 0 to 216 h, and the growth curve was drawn.

### Termination reaction
All PET hydrolysis reactions were terminated by diluting the aqueous solution with 18 mM phosphate buffer (pH 2.5) containing 10% (v/v) DMSO, followed by heat treatment (85 °C, 10 min). All BHET hydrolysis reactions were terminated by diluting the aqueous solution

with 16 mM phosphate buffer (pH 2.5) containing 20% (v/v) DMSO followed by heat treatment (80 °C, 10 min). The supernatant obtained by centrifugation (13,500×*g*, 10 min) was analyzed by HPLC.

## Calculation of the LOD and recovery of HPLC analysis

According to Supplementary Fig. 29, we determined that the LOD of HPLC analysis was 0.029 nmol. The recovery rate was ~100% in our study. After the PET hydrolysis reaction was terminated, there were three locations for the major degradation product MHET. The first location was the supernatant of the reaction buffer, the second was the surface of the PET film where MHET might adsorb, and the last was the inner surface of the test tube where MHET might adsorb as well. Therefore, we measured MHET samples from those three locations. The first sample was obtained by centrifugation. The second and third samples were obtained by washing the PET film and test tube, respectively. After the HPLC measurements, we found that nearly 100% of MHET was in the supernatant of the reaction buffer (Supplementary Fig. 16). This result was consistent with the result from Yoshida's study, which only measured MHET from the supernatant of the reaction buffer.

## Differential scanning calorimetry (DSC) measurements

Differential scanning calorimetry was performed using a DSC214 polyma (NETZSCH, Germany) using ≈6–8 mg of a dry PET sample. A heating rate of 10 °C min$^{-1}$ was applied for the temperature range from −20 to 300 °C. The glass transition temperature ($T_g$), cold crystallization temperature ($T_{cc}$), and melting point ($T_m$) of various PET samples were obtained using the first heating scan. The initial fraction crystallinity $X_0$ was calculated as follows:

$$X_0 = X_\infty + \frac{\triangle H_{cc}}{\triangle H_m^0(T_{cc})} \quad (3)$$

$$X_\infty = \frac{\triangle H_m}{\triangle H_m^0(T_m)} \quad (4)$$

$$\triangle H_m^0(T_{cc}) = \triangle H_m^0(T_m) - \triangle C_p(T_m - T_{cc}) \quad (5)$$

as described before[81,82], where $X_0$ is the initial crystallinity, $X_\infty$ is the complete crystallinity, $\triangle H_m$ is the melting enthalpy, and $\triangle H_{cc}$ is the cold crystallization enthalpy. $\triangle H_m^0(T_m)$ is the melting enthalpy of pure crystalline PET at a melting temperature of 140 J g$^{-1}$[83]. $\triangle H_m^0(T_{cc})$ is the melting enthalpy of pure crystalline PET at the temperature of cold crystallization, which can be calculated according to Eq. (5). $\triangle C_p$ is the difference in the heat capacity of amorphous and crystalline PET of 0.17 J g$^{-1}$ K$^{-1}$[81].

## AlphaFold modeling

In this study, AlphaFold2 was used to predict PETase-linker-GCW51 and HFBI-linker-GCW61. The prediction of complexes was run twice with different random seeds, and ten models were obtained. Beginning with a visual inspection, four models were selected to check the protein structural quality for the side chain conformations using the prime module of Schrödinger 2021-3. Eventually, the one complex with the highest quality score was selected for further optimization with subsequent MD simulations.

## Molecular dynamics simulations

The all-atom MD simulations were performed by Gromacs 2019.6 with CHARMM36 force field[84,85]. The structural models of PETase and GCW51 fusion protein, HFBI and GCW61 fusion protein were generated by AlphaFold2[86]. The 4PET polymer chain and the corresponding parameter files (.top and.itp) were generated following previously published procedures[87]. For the modeling of solution-state PETase, a predocked

PETase/4PET or PETase-GCW51/4PET complex was solvated with TIP3P waters and ions (0.15 M NaCl, total of 164,478 atoms). The Nose–Hoover thermostat (303.15 K) and Parrinello-Rahman isotropic NPT ensembles were adopted with h-bond LINCS constraints. Three independent trajectories were performed (250 ns each) after a short 10 ns equilibrium simulation. To model the membrane-bound PETase, PETase-GCW51 and HFBI-GCW61 were anchored to a bilayer of lipids through glycosyl-phosphatidylinositol modifications at protein N-termini. A typical composition of lipids included 66 molecules of ergosterol, 32 molecules of POPA, 64 molecules of POPC, 60 molecules of POPE, and 96 molecules of POPI to mimic the yeast membrane environment and maintain the bilayer stability (15 nm × 15 nm in the xy dimension)[88,89]. To mimic PET plastic in a possible solvent-swelling state, 20 individual chains of 4PET were added to the system at random positions along with 6 nm thickness of TIP3P waters and ions (0.15 M NaCl), the resulting system contained a total of 343086 atoms. After equilibrium was reached, simulations were performed using three independent trajectories for 50 ns. Representative conformations were extracted from each trajectory by *gmx cluster* for further analysis. Quantitative analysis was also implemented using *gmx rms, rmsf, Rg* tools.

## Molecular docking

Induced fit docking (IFD) experiments were performed using Schrödinger 2021-3, employing the Maestro graphical interface on PETase-Linker-GCW51 generated by AlphaFold2. A PET-tetramer capped at both ends as ethyl esters was prepared and minimized in Maestro using the LigPrep module with the OPLS3e force field[90]. The PETase-Linker-GCW51 structure was prepared in Maestro employing the Protein Preparation Wizard[91]. Protonation states were assigned according to the pKa values and a pH of 7 with PropKa. Finally, the structures were refined through restrained minimization using the OPLS3e force field to within an RMS gradient of 0.1 kcal mol$^{-1}$ Å$^{-1}$. Finally, a Glide redocking step was performed using the extra precision Grid-Based Ligand Docking with Energetics (Glide XP) algorithm. In Glide XP docking, a better correlation between best poses and scores was obtained[73].

## Reporting summary

Further information on research design is available in the Nature Portfolio Reporting Summary linked to this article.

## Data availability

The coordinates for crystal structures of wild-type PETase and PETase-linker have been deposited in the Protein Data Bank (PDB), with the accession codes 8GU5 [https://doi.org/10.2210/pdb8GU5/pdb] and 8GU4 [https://doi.org/10.2210/pdb8GU4/pdb], respectively. The data generated in this study are provided in the Supplementary Information and the Source Data file provided with this paper. Data were also available from the corresponding author upon request. Source data are provided with this paper.

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

## Acknowledgements

We thank the staff from beamlines BL17U1, BL18U1, BL02U1, and BL19U1 at Shanghai Synchrotron Radiation Facility (China). This work was supported by grants from the National Natural Science Foundation of China (No. 31970048) to Z.W., the National Natural Science Foundation of China (No. 61971302) to Yanyan W., the National Natural Science Foundation of China (NO. 82202518) to Y.X., the National Natural Science Foundation of China (NO. 22007071and 22077094) to C.Z.

## Author contributions

Z.W. and Yanyan W. conceived and supervised the project; Z.W., Yanyan W., Yi W., Z.C., Y.X., R.D., and H.Y. designed the experiments; C.Z. and Y.Y. performed the computational simulations; Z.W., Yanyan W., Y.X., Yi W., Z.C., R.D., C.Z., X.S., Y.C., X.W., S.T., H.Z., S.W., and H.Y. analyzed and discussed the data; Z.W., Yanyan W., Yi W., Z.C., R.D., C.Z., and Y.X. wrote the manuscript.

## Competing interests

The authors declare no competing interests.
