## [Peer Review File · Nature Communications]

Biodegradation of Highly Crystallized Poly(ethylene terephthalate) Through Cell Surface Codisplay of Bacterial PETase and HydrophobinREVIEWER COMMENTS

Reviewer #1 (Remarks to the Author):

Title

Efficient Biodegradation of Highly Crystallized 1 Poly (ethylene terephthalate) Through Cell Surface Co-Display of Bacterial PETase and Hydrophobin

Authors

Yi Wei, Zhuozhi Chen, Yunjie Xiao, Yingying Cheng, Xue Wang, Shanwei Tong, Haitao Yang, Yanyan Wang, Zefang Wang

Summary

Wei and co-workers developed a co-display system where the hydrophobin HFBI and the hydrolytic enzyme PETase were exposed on the surface of yeast cells. Despite both the hydrophobin protein and the PETase enzyme are well known in the literature, I believe that this approach is novel and worth considering for publication in a journal with an excellent reputation such as Nature Communications.

This work will be of significance in the polymer biotechnology and plastics recycling fields as a whole cell based biocatalysis process such as the one described in this work that allows the simultaneous regulation of the adsorption and depolymerization step could be of great interest also when applied on other materials.

Comments

- The introduction is complete and cites all relevant literature in the field. The only minor addition I would like the authors to add is the following articles from the Guebitz group in Vienna that reported how thermostable cutinases fused with binding modules have an increased PET films hydrolysis rate and can attack polyester-polyurethane materials:

<https://pubs.acs.org/doi/abs/10.1021/bm400140u>

<https://www.sciencedirect.com/science/article/pii/S0141391016300489>

Reason for adding these references: I believe that this previously described approach is strongly related to the research carried out in this manuscript since the authors mention the lack of a carbohydrate binding module in the investigated enzyme.

- I suggest adding a brief 2-3 sentences introductory paragraph to the R&D section as this part starts directly citing data contained in figures and in the supplementary information file.

- Would be nice to have a figure in the supplementary information showing the hcPET degradation extent as %.

The work supports the conclusions and claims, and I believe that no additional evidence is needed. I did not notice any flaws in the data analysis, interpretation, and conclusions. The methodology and statistical analysis are sound and all details to allow reproducibility were in my opinion included.

I recommend publication of this nice piece of work after the few minor revisions listed above.

Reviewer #2 (Remarks to the Author):

This manuscript describes a cell surface co-display system introduced PETase and hydrophobin (HFBI) in *Pichia pastoris* from the perspective of two steps: adsorption and degradation for efficient PET decomposition in enzymatic degradation process. The authors confirmed that the introduction of this system significantly improved PET decomposition compared to purified PETase through the introduction of this system, and demonstrated that this co-display system has the advantages of thermal-stability, chemical-stability, and substrate-absorption. However, since the effect of hydrophobin on the enhancement of PET hydrolysis by PETase has already been demonstrated in several published papers, novelty of this concept is not high.

Considering the actual application of this co-display system, I wonder about the maximum PET degradation capacity of this system in long term reaction beyond understanding the degree of PET

degradation activity a specific time period. (The maximum decomposition rate...). It will be the result of comprehensive consideration of various factors (effect of cell recycling time, improvement accessibility on substrate, enzyme activity, thermal-stability, chemical-stability and so on...), this will help to evaluate the practical of PET biodegradation technology compared to PET biodegradation system that exist to date.

In the enzymatic assay for PET, all PET degradation activity data were presented as turnover rate and turnover rate was calculated by the amount (moles) of mono (2-hydroxyethyl) terephthalic acid (MHET) in this manuscript. But, in fact, the released PET decomposition products using PETase are not only MHET, but also terephthalic acid (TPA) and bis(2-hydroxyethyl)- terephthalic acid (BHET). In order to compare the enzymatic activity, it may be accurate to consider the degree of hydrolysis of PET as the sum of TPA and MHET as well as the turnover rate of MHET. Was TPA present in trace amounts in all enzymatic assay? It might be negligible in case of BHET.

More specific comments:

1. The authors should explain the structural difference between wildtype PETase and linker attached PETase. While the protein was folded, the linker could affect structural characteristics and protein activity as well as protein stability. It would be better to show the PET degradability with the surface-displaying system without hydrophobin.
2. In line 112, the authors mentioned, "For PETase, what is circled in the structure is the active-site that is much broader than those of the other PET hydrolases, which may explain why PETase can accommodate a large substrate like PET". However, it is controversial since other well-known PET hydrolases show the highest PET degradability and protein stability at 70°C, while PETase shows the highest PET degradability at around 30°C. Thus, the temperature condition should be noted in this sentence.
3. In line 119, the authors insisted, "Theoretically, our co-display system should have dual functions that mimic the two-step process of PET degradation." However, the information on the conformational fitting of PETase and high crystalline PET was not described. Hydrophobin-co-displayed PETase could increase the cell adsorption on hcPET due to its increased hydrophobicity. However, the precise interaction between linker-attached PETase and hcPET was not elucidated. Thus, the data related to the interaction between linker-attached PETase and hcPET should be demonstrated to insist this system mimic the two-step process of PET degradation; (1) The interaction between PETase and hcPET and (2) hydrolysis of hcPET by PETase.
4. In line 127, fluorescent signals were first coming out without any background description. The process to construct fluorescence combined systems and the meaning of this system should be discussed.
5. In line 142 and Supplementary Fig.3, the authors showed "the liberated products" of displayed PETase and that of the native PETase and insisted that the ratio between TPA, MHET and BHET was almost the same. However, the concentration of liberated products should be discussed to compare the activity of the native PETase and the displayed PETase.
6. In a similar manner with comment 15, the concentration of native PETase and displayed PETase used in the experiments was not notified. Since the result of PET degradation was largely dependent on the concentration of PETase and the conditions of reaction, a detailed description should be demonstrated. Also, the soluble and non-soluble ration of PETase expressed in the strain should be described to clarify the PETase was activated or not as an enzyme.
7. The PET degradability of hydrophobin-displayed cells should be examined since the enzymes from yeast cells could degrade PET, which can also increase the amount of PET degradation.
8. In Figure 4 and line 252, microscopic observation was performed to examine the effect of the co-displayed system. The number of test sets for each model should be expanded and analyzed statistically to increase the data credibility.
9. In Supplementary Figure 2, Western blot analysis of GS115/PETase-HFBI was shown. The size of proteins should be clearly notified with a protein marker. Moreover, whether the protein is soluble or aggregated should also be notified to clarify that this system is well constructed without protein aggregation as the authors designed.
10. It would be better to mention the way to measure the crystallinity of hcPET at each temperature in which experiments were performed since crystallinity is an important factor in PET degradation. PETase from *Ideonella sakaiensis* shows a low activity on hcPET due to its limited activity at high

temperatures, while PET shows lower crystallinity at high temperatures, which makes hydrolysis easier.

11. Overall, I recommend showing the cell growth profile and the corresponding protein concentration profile for each control, PETase-displayed, HbFI-displayed strain, and HbFI-PETase-co-displayed strains to clarify the PET degrading mechanisms in this system developed in this study.

12. In addition, I recommend prolonging the incubation time to measure the stability of the newly designed whole-cell biocatalyst. In this study, in vitro PET degradation test was mainly performed in 50 mM glycine-NaOH (pH 9.0) at 30 °C for 18 h. The cultivation time is relatively short to compare the whole PET degrading mechanisms while the native PETase shows longer stability than 18h.

13. Lastly, I recommend performing in vivo one-step PET degradation using this system to maximize the advantages of the whole-cell biocatalyst. This attempt would make this study more competitive to reach industrial scale-up by suggesting the possibility to assimilate PET and recycle PET to other valuable products with greatly high efficiency using cell factories.

14. Please check typos/errors throughout the manuscript: many of them including Results and discussion line 120, mimic should be mimic

Methods line 449, The turnover rate was calculated should be The turnover rate was calculated

Reviewer #3 (Remarks to the Author):

This work described very an interesting work about co-displaying hydrophobin and PETase on the surface of *Pichia pastoris* for enhancing degradation of PETase. Firstly, the co-displaying hydrophobin HbFI with PETase on yeast cells surface was confirmed by several technique such as microscopic observation of florescent stained of yeast surface, number of cells adhesion on PET surface, contact angle measurement, MATH. My main concern of this work is for saying that co-display cells exhibit much higher turnover rates than those of native PETase or PETase displaying cells. The turnover rate was calculated by normalizing the mole of MHET produced per degradation time to the moles of PETase present in the experiments. Protein concentration of PETase displayed was normalized for the comparison of turnover rate. Therefore, the accurate determination of the amount of PETase displayed on yeast surface is very important for calculating the turnover rate. The amount of displayed PETase was determined by Western blot using a gray-scale calibration curve as shown in Figure S9 (a). The question is how can you sure that displayed PETase on yeast surface was completely removed from surface for SDS-PAGE Western blot analysis. In the material and method section, there are no description about how the co-displayed or displayed PETase was removed from yeast surface for Western blot analysis. If less amount of displayed PETase was determined, then it possible leads to a higher turnover number.

The optimization of the degradation system Fig. 3 a is better also compared with PETase-displayed cells and the PETase concentration per cell should be included for comparison. For example 10 displayed PETase per cell compared with 1 per cell, based on the same amount of PETase for turnover, the case with 1 displayed PETase per cell should be more efficient for degradation because the 1 per cell case can have about 10 spots for degradation on PET surface.

Besides, in addition to the enhanced adhesion to PET surface that leads to the enhanced turnover rate. The mechanism for degradation a PET solid surface by a catalytic microparticles (PETase-displayed yeast cells) should be proposed to explain the enhanced adhesion can promote the degradation.

Point-to-point Response to Reviewers' Comments

We thank all reviewers for their thoughtful and thorough remarks. They are addressed in detail below, and we believe the changes we have made in response have significantly improved the manuscript.

Reviewer #1 (Remarks to the Author):

Title

Efficient Biodegradation of Highly Crystallized 1 Poly (ethylene terephthalate) Through Cell Surface Co-Display of Bacterial PETase and Hydrophobin

Authors

Yi Wei, Zhuozhi Chen, Yunjie Xiao, Yingying Cheng, Xue Wang, Shanwei Tong, Haitao Yang, Yanyan Wang, Zefang Wang

Summary

Wei and co-workers developed a co-display system where the hydrophobin HFBI and the hydrolytic enzyme PETase were exposed on the surface of yeast cells. Despite both the hydrophobin protein and the PETase enzyme are well known in the literature, I believe that this approach is novel and worth considering for publication in a journal with an excellent reputation such as Nature Communications.

This work will be of significance in the polymer biotechnology and plastics recycling fields as a whole cell based biocatalysis process such as the one described in this work that allows the simultaneous regulation of the adsorption and depolymerization step could be of great interest also when applied on other materials.

We thank the reviewer for the appreciation of our work, your encouraging comment is greatly appreciated.

Comments

- The introduction is complete and cites all relevant literature in the field. The only minor addition I would like the authors to add is the following articles from the Guebitz group in Vienna that reported how thermostable cutinases fused with binding modules have an increased PET films hydrolysis rate and can attack polyester-polyurethane materials:

<https://pubs.acs.org/doi/abs/10.1021/bm400140u>

<https://www.sciencedirect.com/science/article/pii/S0141391016300489>

Reason for adding these references: I believe that this previously described approach is strongly related to the research carried out in this manuscript since the authors mention the lack of a carbohydrate binding module in the investigated enzyme.

Thanks for the reviewer's helpful comment. These two articles are excellent and highly related to the research content of our manuscript. In the first article, two different binding modules were respectively fused with the Thc_Cut1 to improve sorption and hydrolysis (*Biomacromolecules*, 2013). The results showed that the activity of both fused proteins toward

PET was increased compared with the native enzyme. In another article, a polyamidase from *Nocardia farcinica* (PA) was fused to a polymer binding module from a polyhydroxyalkanoate depolymerase from *Alcaligenes faecalis* (PA_PBM) (*Polym. Degrad. Stab.*, 2016). And the PA_PBM fusion enzyme was up to 4 times more active on the polymer when compared with the native enzyme, confirming the relevance of enzyme adsorption for efficient hydrolysis. This provides favorable support for us to use HFBI to enhance the adsorption of PET and hence improve the degradation activity.

As suggested, we have cited the two articles from Prof. Guebitz's group in the revised manuscript. The added content was listed as follows: Several binding modules have been reported to have the ability to enhance the enzymatic degradation of polymers by increasing enzyme adsorption (*Biomacromolecules*, 2013; *Polym. Degrad. Stab.*, 2016). Lines 78-80.

- I suggest adding a brief 2-3 sentences introductory paragraph to the R&D section as this part starts directly citing data contained in figures and in the supplementary information file.

Thanks for the helpful comment. We have added three sentences accordingly at the beginning of the R&D section in the revised manuscript. The added content was listed as follows:

“In our co-display system, hydrophobin HFBI and PETase are supposed to play different roles based on their unique protein structures. HFBI is thought to regulate the adsorption of yeast cells on the substrate PET. PETase is responsible for degrading the substrate PET. Fig. 1a shows the structures...”. Lines 115-117.

- Would be nice to have a figure in the supplementary information showing the hcPET degradation extent as %.

Thank you for the suggestion. We have added a figure (Figure S24) in the supplementary information showing the hcPET degradation extent as 3.0% and the lcPET degradation extent as 55%.

Figure S24. The degradation rates of hcPET and lcPET within 18 h at 30 °C were about 3.0% and 55%, respectively.

The work supports the conclusions and claims, and I believe that no additional evidence is needed. I did not notice any flaws in the data analysis, interpretation, and conclusions. The methodology and statistical analysis are sound and all details to allow reproducibility were in my opinion included. I recommend publication of this nice piece of work after the few minor revisions listed above.

We thank the reviewer for the appreciation of our work, your encouraging comment is greatly

appreciated. We have carefully revised the manuscript in response to your comments.

Reviewer 2

This manuscript describes a cell surface co-display system introduced PETase and hydrophobin (HFBI) in *Pichia pastoris* from the perspective of two steps: adsorption and degradation for efficient PET decomposition in enzymatic degradation process. The authors confirmed that the introduction of this system significantly improved PET decomposition compared to purified PETase through the introduction of this system, and demonstrated that this co-display system has the advantages of thermal-stability, chemical-stability, and substrate-absorption. However, since the effect of hydrophobin on the enhancement of PET hydrolysis by PETase has already been demonstrated in several published papers, novelty of this concept is not high.

We thank the reviewer for careful reading of our manuscript. We agreed with the reviewer that hydrophobin has been reported to increase the enzymatic degradation performance of PETase by pretreatment itself or working as part of a fusion protein. The speculated mechanism underlying those observations is hydrophobin can wet the surface of PET and leads PETase easier to contact and attack the hydrophobin-modified PET surface. (*Appl. Environ. Microbiol.*, 2015; *Appl Biochem Biotechnol*, 2021; *Int. J. Biol. Macromol.*, 2021).

In our co-display system, we proved that displayed hydrophobin HFBI could work as an adhesive unit to increase the immobilization of the yeast cells (not the enzyme) onto the PET surface by altering the surface hydrophobicity of the yeast cells. In the newly added Figure 5, MD simulation experiment also confirmed that the PET molecule bound displayed HFBI firstly before it was hydrolyzed in the active pocket of PETase. In the pre-equilibrated system (0 ns), randomly distributed 4PET did not interact with the active center of PETase nor hydrophobic region of HFBI (Fig. 5a and Fig. S26). As time went on, 4PET gradually aggregated near the hydrophobic amino acids of HFBI at about 40-50 ns. At about 60-70 ns, 4PET completely attached to the hydrophobic patch of HFBI mainly through interaction of hydrophobic residues and remain mostly bound for the rest of the trajectory. During this period, 4PET began to gather near the active center of PETase, eventually a number of 4PET molecules were gathered at about 100 ns. The calculation of the distances between 4PET and the other two proteins collaborated with the possible role of HFBI in local enrichment of PETs (Figure S27). From the simulated trajectories, we postulated that our co-display system complete PET degradation through two-step processes of adsorption and hydrolysis, wherein the hydrophobin HFBI favorably binds to the PET chains and grabs them firmly (Fig. 5b), and then PETase interacts with PET and exerts hydrolysis function (Fig. 5c).

In summary, we performed cell immobilization other than performing enzyme immobilization in previous studies by using hydrophobin. From this point of view, our research is still innovative in the application of hydrophobin as a cell-adhesive unit.

Figure 5. Two-step process of degradation of PET by the co-display system

Considering the actual application of this co-display system, I wonder about the maximum PET degradation capacity of this system in long term reaction beyond understanding the degree of PET degradation activity a specific time period. (The maximum decomposition rate...). It will be the result of comprehensive consideration of various factors (effect of cell recycling time, improvement accessibility on substrate, enzyme activity, thermal-stability, chemical-stability and so on...), this will help to evaluate the practical of PET biodegradation technology compared to PET biodegradation system that exist to date.

Thanks for the reviewer's nice comments. In order to evaluate the maximum PET degradation capacity of this co-display system in long term reaction. We used the co-display system to degrade hcPET for a long time, and the PETase display system and the wild type PETase were used as control. Detailed data are put in the response to the 12th specific comment below.

In the enzymatic assay for PET, all PET degradation activity data were presented as turnover rate and turnover rate was calculated by the amount (moles) of mono (2-hydroxyethyl) terephthalic acid (MHET) in this manuscript. But, in fact, the released PET decomposition products using PETase are not only MHET, but also terephthalic acid (TPA) and bis(2-hydroxyethyl)- terephthalic acid (BHET). In order to compare the enzymatic activity, it may be accurate to consider the degree of hydrolysis of PET as the sum of TPA and MHET as well as the turnover rate of MHET. Was TPA present in trace amounts in all enzymatic assay? It might be negligible in case of BHET.

Thanks for the helpful comment. The total amounts of released products including MHET, TPA and BHET were calculated in the revised manuscript. Detailed data are put in the

response to the 5th specific comment below.

More specific comments:

1. The authors should explain the structural difference between wildtype PETase and linker attached PETase. While the protein was folded, the linker could affect structural characteristics and protein activity as well as protein stability.

Thanks for the nice comments. Some linkers do affect the structural characteristics and activity of proteins (*Appl. Biochem. Biotechnol.*, 2016; *Mol. Biosyst.*, 2017; *Adv. Drug Deliv. Rev.*, 2013). In fusion protein design strategies, the flexibility and length of linkers are important parameters affecting the bioactivity of multifunctional proteins. Researchers generally design linkers as flexible linkers or rigid linkers. Wherein, flexible linkers are usually applied when the joined domains require a certain degree of movement or interaction (*J. Mol. Biol.*, 1990; *Adv. Drug Deliv. Rev.*, 2013). The most commonly used flexible linkers have sequences primarily consisting of stretches of Gly and Ser residues (“GS” linker), and the Ser could reduce the unfavorable interaction between the linker and the protein moieties. “GS” linker can increase spatial separation between domains. Therefore, we chose the “GS” linker to construct the fusion protein of PETase-linker-GCW51 (Figure 1b).

To explore the structural difference between wild type PETase and linker attached PETase, the crystal structures of those two proteins were determined at 2.0 Å and 1.5 Å respectively (Figure 1c-d, Figure S1 and Table S1). We compared the structure of wild type PETase and linker attached PETase, and found that the tertiary structures of these two proteins were quite similar with an overall RMSD of 0.352 Å. In the structure of linker attached PETase, we can only see a part of Linker (6 of amino acids) in the form of irregular structure, indicating that (GGGGS)₂ linker is flexible. This result is consistent with previous reports that GGGGS sequence is the most widely used flexible linker which can achieve appropriate separation of the functional domains (*Adv Drug Deliv Rev.*, 2013). The catalytic centers of those two proteins were compared as well. As shown in Figure 1d, two active-site pockets were almost identical. Those above results revealed that C-terminal fused linker did not pose obvious structures changes to the PETase.

The overall structure of the wild-type PETase (a) and PETase-linker (b). (c) Comparison of three-dimensional spatial structures of wild-type PETase and PETase-linker. (d) Catalytic triad comparison of wild-type PETase with PETase-linker.

In order to validate our structural findings, we also performed Molecular dynamics (MD) studies for the co-displayed PETase and HFBI. In the simulation, (GGGGS)₂ linker shows high flexibility and no unique structure when linker attached PETase is displayed by the yeast cell-

wall protein GCW51. Moreover, this flexible linker separates PETase from GCW51 in space, meaning that the two functional units do not affect each other (Figure S2).

Figure S2. Molecular dynamics (MD) studies for the co-displayed PETase and HFBI. Model for PETase-Linker-GCW51 (a) and HFBI-Linker-GCW61 (b) were built with AlphaFold. The Ramachandran plot of the PETase-Linker-GCW51 (c) and HFBI-Linker-GCW61 (d). Molecular dynamics simulations of Display system in 0 ns (e) and 50 ns (f) and Co-display system in 0 ns (g) and 50 ns (h).

It would be better to show the PET degradability with the surface-displaying system without hydrophobin.

Thank you for the suggestion. In the revised manuscript, we added a new control for our co-display system. In this new co-display system, only hydrophobin was removed and other parts remained unchanged. Figure S13a shows that both liker-GCW61 and PETase were successfully displayed on the surface of yeast cells. Then the enzymatic degradation performance against hcPET and lcPET was tested. We found that co-display system without hydrophobin exhibited greatly reduced degradation ability towards no matter hcPET or lcPET compared with normal co-display system (Figure S13b-c).

Figure S13. Fluorescence microscopy and the enzyme activity of Hydrophobin removed display system.

We also found that its adsorption capacity on the PET substrate is obviously reduced, about a half of the original (Figure S14). These results clearly revealed that the adhesive unit HFBI was particularly important for the high degradation capacity to PET of the co-display system.

Figure S14. The adsorption of the yeast cells of Hydrophobin removed display system.

2. In line 112, the authors mentioned, "For PETase, what is circled in the structure is the active-site that is much broader than those of the other PET hydrolases, which may explain why PETase can accommodate a large substrate like PET. However, it is controversial since other well-known PET hydrolases show the highest PET degradability and protein stability at 70 °C, while PETase shows the highest PET degradability at around 30 °C. Thus, the temperature condition should be noted in this sentence.

Thank you for the suggestion. The temperature condition has been added in the revised sentence. "For PETase, what is circled in the structure is the active-site that is much broader than those of the other PET hydrolases, which may explain why PETase can accommodate a large substrate like PET at moderate temperatures."

3. In line 119, the authors insisted, "Theoretically, our co-display system should have dual functions that mimic the two-step process of PET degradation." However, the information on the *conformational fitting* (<https://doi.org/10.1038/s41467-019-13492-9>, Conformational fitting of a flexible oligomeric substrate does not explain the enzymatic PET degradation) of PETase and high crystalline PET was not described. Hydrophobin-co-displayed PETase could increase the cell adsorption on hcPET due to its increased hydrophobicity. However, the precise interaction between linker-attached PETase and hcPET was not elucidated. Thus, the data related to the

interaction between linker-attached PETase and hcPET should be demonstrated to insist this system mimic the two-step process of PET degradation; (1) The interaction between PETase and hcPET and (2) hydrolysis of hcPET by PETase.

Thanks for this valuable comment. The reviewer asked a fundamental question about how the displayed PETase interacts with its substrate PET. This is a very challenging question. We discussed this issue with many researchers in the field, and finally came to a conclusion. That is, according to the current technical means, it is difficult to use classical biochemical and biophysical methods to clarify this question. Molecular dynamics simulation seems to be a relatively reliable solution at present. This method has been adopted by several researchers to investigate the interaction of wild-type and mutant PETase and its substrate PET in a free state (*ACS Catal.* 2022; *ACS Catal.* 2021; *Proc Natl Acad Sci U S A*, 2018; *Proteins*, 2021; *J. Chem. Inf. Model.*, 2021; *ACS Sustain Chem Eng.*, 2021).

Figure 5. Two-step process of degradation of PET by the co-display system

We performed molecular dynamics simulation analysis. In the pre-equilibrated system (0 ns), randomly distributed 4PET did not interact with the active center of PETase nor hydrophobic region of HFBI (Fig. 5a and Fig. S26). As time went on, 4PET gradually aggregated near the hydrophobic amino acids of HFBI at about 40-50 ns. At about 60-70 ns, 4PET completely attached to the hydrophobic patch of HFBI mainly through interaction of hydrophobic residues and remain mostly bound for the rest of the trajectory. During this period, 4PET began to gather near the active center of PETase, eventually a number of 4PET molecules were gathered at about 100 ns. The calculation of the distances between 4PET and the other two proteins collaborated with the possible role of HFBI in local enrichment of PETs (Figure S27). From the simulated trajectories, we postulated that our co-display system

complete PET degradation through two-step processes of adsorption and hydrolysis, wherein the hydrophobin HFBI favorably binds to the PET chains and grabs them firmly (Fig. 5b), and then PETase interacts with PET and exerts hydrolysis function (Fig. 5c).

Next, in order to obtain a more accurate interaction between PETase-linker-GCW51 and hcPET, and to elucidate whether PETase-linker-GCW51 follows a conformational selection or induced-fit mechanism, we performed an induced fit molecular docking analysis of PETase-linker-GCW51 with a PET tetramer representing the polymer substrate (4PET). The five highest-scoring docking poses were subjected to analysis with respect to spatial arrangement of the residues of the catalytic triad and the conformation of the reactive part of the oligomer substrate (*ACS Catal.*, 2022). And we found the best-predicted docking pose taking productivity of the catalytic triad into account is shown in Figure 5c. The reacting carbonyl carbon of the substrate was bound with the scissile ethylene glycol moiety in the canonical gauche conformation (Ψ_{gauche}) with the chain twisted. This was consistent with the way in which wild-type PETase binds to 4PET (*ACS Catal.*, 2022).

Figure S28. Schematic diagram of hydrolysis of hcPET by Co-display system.

In order to more clearly understand the two-step process of hcPET degradation by the co-display system, we mapped the mechanism as shown in Figure S28. Firstly, due to the presence of HFBI, co-display cells quickly adsorbed to the hcPET surface, and the adsorption rate on the hcPET surface was close to 100% (Figure S10 and S11). This could also be further confirmed by the results of scanning electron microscopy. Whether it was lcPET or hcPET, the corrosion spots observed on the surface basically cover the surface, while wild-type PETase only cuts in part of the PET area (Figure 3, S15, S20). Secondly, PETase contacts the surface of high crystallinity PET, and then hydrolyzes the PET chains, thus achieving the effect of

efficient hydrolysis of high crystallinity PET.

4. In line 127, fluorescent signals were first coming out without any background description. The process to construct fluorescence combined systems and the meaning of this system should be discussed.

Thanks for the helpful comment. In our study, immunofluorescence was used to detect the location of HFBI and PETase expressed in the yeast cell. In theory, both proteins have anchor proteins, so we expect them to be detected on the cell surface. Flag tag was introduced to the N terminal of PETase to response the detecting of anti-flag antibody. For the displayed HFBI, monoclonal antibody against HFBI was used in the immunofluorescence analysis. By doing immunofluorescence, it was confirmed that both HFBI and PETase were anchored on the cell surface of *P. pastoris*. The detailed molecular construction and immunofluorescence staining steps can be found in the Methods section.

5. In line 142 and Supplementary Fig.3, the authors showed “the liberated products” of displayed PETase and that of the native PETase and insisted that the ratio between TPA, MHET and BHET was almost the same. However, the concentration of liberated products should be discussed to compare the activity of the native PETase and the displayed PETase.

Thanks for the helpful comment. The total amounts of released products including MHET, TPA and BHET were re-calculated in the revised manuscript. All re-calculated product data are summarized in following Table. It was clear that the displayed PETase greatly elevated the total amounts of released products at the similar operation condition, suggesting the high enzymatic activity of the displayed PETase compared with the native PETase.

Degradation system	PET film (crystallinity)	BHET (μM)	TPA (μM)	MHET (μM)	Total products (μM)	Turnover rate (sec^{-1})
Co-display	45%	14.23	42.69	455.38	515.15	2146.5×10^{-5}
	6%	10270.03	28800.74	302910.09	341980.86	2909.3×10^{-2}
Display	45%	1.64	4.58	48.46	54.68	227.9×10^{-5}
	6%	857.88	2405.72	25300.64	28563.92	243.0×10^{-2}
PETase	45%	0.04	0.11	1.41	1.56	6.5×10^{-5}
	6%	15.53	46.89	595.85	658.26	5.6×10^{-2}

6. In a similar manner with comment 15, the concentration of native PETase and displayed PETase used in the experiments was not notified. Since the result of PET degradation was largely dependent on the concentration of PETase and the conditions of reaction, a detailed description should be demonstrated. Also, the soluble and non-soluble ration of PETase expressed in the strain should be described to clarify the PETase was activated or not as an enzyme.

Thanks for the helpful comment. We added a detailed description about the concentration of native PETase and displayed PETase used in the experiments. The corresponding details are presented below :

To evaluate the hydrolytic activity of PETase and the co-displayed recombinant *P. pastoris*, the hcPET film (Good Fellow, crystallinity, 45%, thickness, 0.175 mm, diameter, 6 mm) and the lcPET film (Goodfellow Cambridge, PET-GF, crystallinity of 6.3%, 0.25 mm thick, diameter, 6 mm) were used as the substrates for degradation assays with the purified PETase enzyme and the displayed system. Before the reaction, the PET film were separately soaked in 0.5% Triton X- 100, 10mM Na₂CO₃ and distilled water, each remained 30 min at 50 °C for 550 rpm, and then air-dried for the reaction. Subsequently, the PET film was put into a tube with 300 μL buffer containing 50 mM glycine-NaOH (pH 9.0) for 18 h at 30 °C with 370 nM purified enzyme and corresponding yeast cells. To optimize induction time, the purified PETase and GS115/PETase-HFBI displayed yeast cells were induced for 0 h, 24 h, 48 h, 72 h, and 96 h for enzyme activity assay. Followed by removing the enzyme-treated PET film from the reaction mixture, the enzyme reaction was terminated by heating at 85 °C for 10 min. The reaction mixture samples were then centrifuged at 12,000 g for 5 min. The supernatant of each sample was further analysed by high-performance liquid chromatography (HPLC) for quantifying PET monomers released from the PET depolymerization. To compare the PET-hydrolytic activity of the displayed system with WT PETase across a range of pH (2.0–10.0) at 20 and 50 °C, a similar experimental setup was used.

To quantify the displayed PETase concentration, we used Western blot for quantitative analysis. First, we used PETase for Western blot grayscale analysis and plotted a standard curve, as shown in Figure S6a-b. Subsequently, we took 2×10^7 yeast cells after 48 h induction for SDS-PAGE electrophoresis and quantitative analysis by Western blot, as shown in Fig. S6c. According to the standard curve, for the Display system, the PETase expression amount of 2×10^7 cells is equivalent to 151.2 ng. And for the Co-display system, the PETase expression amount of 2×10^7 cells is equivalent to 35.2 ng. PET degradation experiments in the manuscript were carried out by taking corresponding multiples of 2×10^7 yeast cells, and the concentration of PETase was calculated according to the proportion.

Figure S6. Displayed PETase quantitative analysis.

In addition, to determine the soluble and non-soluble ration of PETase expressed in the yeast strain, we performed Western blot analysis on whole cells, cell walls and protoplasts (See Materials and Methods for detailed extraction methods of PETase) according to the literature (*Enzyme Microb Technol.*, 2020; *Biotechnol Lett*, 2011), and the results were shown in Figure S4. For co-display system, compared with the expression of PETase in the whole cell, the protein expression on the cell wall accounted for $101 \pm 4.31\%$, while the protoplast samples had almost no protein. The results of the display system were similar to those of the co-display system, which indicated that PETase were all displayed on the cell surface, and the

degradation of PET was not in the form of free enzymes.

Figure S4. Western blot analysis on whole cells, cell walls and protoplasts of the soluble and non-soluble ration of PETase expressed.

7. The PET degradability of hydrophobin-displayed cells should be examined since the enzymes from yeast cells could degrade PET, which can also increase the amount of PET degradation.

Thanks for the reviewer's nice comments. In order to verify whether the enzymes in the cells have the activity of degrading PET, we reconstructed some new displayed systems. This co-display system showed high enzyme activity, which should theoretically be caused by PETase. Therefore, we constructed co-display systems that mutant the PETase active center (S160A) or removed PETase. In addition, to prove that the enzyme activity is not generated by hydrophobins, we constructed the HFBI single displayed system.

Figure S8. Immunofluorescence and enzyme activity analysis of different displayed systems.

Those displayed systems proved by immunofluorescence that the proteins displayed were expressed on the cell surface, as shown in Figure S8 a-c. Subsequently, we performed enzyme activity tests using the different displayed systems, and the results were shown in Figure S8 d-e. The results showed that all the displayed systems constructed above did not exhibit PET degradation activity, indicating that the PETase itself plays an enzymatic function rather than hydrophobin or the enzymes from yeast cells.

8. In Figure 4 and line 252, microscopic observation was performed to examine the effect of the co-displayed system. The number of test sets for each model should be expanded and analyzed statistically to increase the data credibility.

Figure S16. Visualization of hcPET film degradation.

We thank the reviewer for pointing out this issue. We increased the number of tests per model, as shown in Figure S16 and Figure S21. It can be seen from the figures that both the hcPET and lcPET results were the same as before, indicating that the results were statistically significant. In addition, it can be seen from the SEM results that the PET films degraded by the co-display system were full of round pits with a size of about 5 microns, which was basically consistent with the size of yeast cells. It shows that yeast cells adsorb on the PET surface and then exert enzyme activity, as seen in the adsorption experiment (Figure S10 and S11). While the purified PETase has small corrosion spots locally and was very small. Which indicated that the co-display system could efficiently degrade lcPET and hcPET.

Figure S21. Visualization of lcPET film degradation.

9. In Supplementary Figure 2, Western blot analysis of GS115/PETase-HFBI was shown. The size of proteins should be clearly notified with a protein marker. Moreover, whether the protein is soluble or aggregated should also be notified to clarify that this system is well constructed without protein aggregation as the authors designed.

Thank you for the suggestion. We remake supplementary Figure 2, as shown in new Figure S3, where the protein marker was added. From the revised figure, it was clear that the molecular weights of PETase and hydrophobin HFBI displayed on the yeast cell surface were consistent with the theoretical molecular weights. In addition, we performed Western blot analysis on whole cells, cell walls and protoplasts, and the results were shown in Figure S4. For co-display system, compared with the expression of PETase in the whole cell, the protein expression on the cell wall accounted for $101 \pm 4.31\%$, while the protoplast samples had almost no protein. The results of the display system were similar to those of the co-display system, which indicated that PETase was all displayed on the cell surface and no PETase was soluble in the cells.

Figure S3. Western blot analysis of GS115/PETase-HFBI.

Figure S4. Western blot analysis on whole cells, cell walls and protoplasts of the soluble and non-soluble ration of PETase expressed.

10. It would be better to mention the way to measure the crystallinity of hcPET at each temperature in which experiments were performed since crystallinity is an important factor in PET degradation. PETase from *Ideonella sakaiensis* shows a low activity on hcPET due to its limited activity at high temperatures, while PET shows lower crystallinity at high temperatures, which makes hydrolysis easier.

Thank you for the suggestion. Differential scanning calorimetry was performed using a DSC214 polyna (NETZSCH, Germany) using $\approx 6-8$ mg of a dry PET sample. A heating rate of $10\text{ }^{\circ}\text{C min}^{-1}$ was applied for the temperature range from -20 to $300\text{ }^{\circ}\text{C}$. The glass transition temperature (T_g), the cold crystallization temperature (T_{cc}), and the melting point (T_m) of various PET samples were obtained using the first heating scan. The initial fraction crystallinity X_0 was calculated according to

$$x_0 = x_{\infty} + \frac{\Delta H_{cc}}{\Delta H_m^0(T_{cc})} \quad (1)$$

$$x_{\infty} = \frac{\Delta H_m}{\Delta H_m^0(T_m)} \quad (2)$$

$$\Delta H_m^0(T_{cc}) = \Delta H_m^0(T_m) - \Delta C_p(T_m - T_{cc}) \quad (3)$$

as described before (*J. Mater. Res.* 2011; *Adv. Sci.*, 2019), where X_0 is the initial crystallinity, X_∞ is the complete crystallinity, ΔH_m is the melting enthalpy, and ΔH_{cc} is the cold crystallization enthalpy. $\Delta H_m^0(T_m)$ is the melting enthalpy of pure crystalline PET at the melting temperature of 140 J g^{-1} (*J. Polym. Sci., Polym. Phys. Ed.*, 1978). $\Delta H_m^0(T_{cc})$ is the melting enthalpy of pure crystalline PET at the temperature of cold crystallization, which can be calculated according to Equation (3). ΔC_p is the difference of the heat capacity of amorphous and crystalline PET of $0.17 \text{ J g}^{-1} \text{ K}^{-1}$ (*J. Mater. Res.* 2011; *Adv. Sci.*, 2019).

We tested the crystallinity of lcPET and hcPET before and after degradation at different temperatures using the co-display system, as shown in Figure S22. For lcPET, it can be seen that the crystallinity increases with the raise of the degradation rate, which indicated that PETase hydrolysis amorphous PET in the co-display system. However, the crystallinity of the PET film treated with the system without enzyme at different temperatures did not change much compared with the control group, indicating that the range of 20-50 °C had little effect on the crystallinity of PET. The change of crystallinity observed in the experiment was mainly caused by enzyme digestion. The same results can also be observed in hcPET samples, except that the crystallinity decreases with the increase of PET degradation, which indicates that the co-display system has no selectivity for PET hydrolysis. At the same time, the change in temperature did not affect the crystallinity of hcPET film, indicating that the change in crystallinity of PET was caused by PETase.

Figure S22. The crystallinity of lcPET and hcPET were determined at 20, 30, 40, and 50 °C experimental temperatures.

11. Overall, I recommend showing the cell growth profile and the corresponding protein concentration profile for each control, PETase-displayed, HBF1-displayed strain, and HBF1-PETase-co-displayed strains to clarify the PET degrading mechanisms in this system developed in this study.

Thank you for the suggestion. We monitored the cell growth curve of Display PETase, Co-display PETase and HBF1, Display HBF1, and GS115, and found that all of the different yeast cells grew normally, with an OD₆₀₀ value of about 9 at 96 h. Meanwhile, we used Western blot analysis to quantitatively analyze PETase expressed at different times, as shown in the following figure. It can be shown that the protein concentration of PETase gradually rises with the increase of time. In addition, the protein concentration of PETase in Co-display system cells was significantly lower than that of the Display system, which further indicated that the hydrophobin HBF1 and PETase are co-displayed on the cell surface. Concurrently, this is also the reason why the turnover rate of the co-display system is far higher than that of the single display system.

The cell growth profile and the corresponding protein concentration profile for different displayed system.

12. In addition, I recommend prolonging the incubation time to measure the stability of the newly designed whole-cell biocatalyst. In this study, in vitro PET degradation test was mainly performed in 50 mM glycine-NaOH (pH 9.0) at 30°C for 18 h. The cultivation time is relatively short to compare the whole PET degrading mechanisms while the native PETase shows longer stability than 18h.

Thanks for the reviewer's nice comments. In order to evaluate measure the stability of the newly designed whole-cell biocatalyst and the maximum PET degradation capacity of this co-display system in long term reaction. We used the co-display system to degrade hcPET for a long time, and the PETase display system and purified PETase were used as control. As shown in Figure 4e-f, the relative degradation rate of PET in both the Co-display system and the Display system increased with time, and reached the maximum around the ninth day, which was 10.9% and 1.2% respectively. However, the degradation activity of wild-type PETase for hcPET was very low, and the degradation rate was only 0.003%. These results shown that the establishment of the co-display system increases the stability of PETase.

Figure 4e-f. The co-display system degraded hcPET for a long time period.

13. Lastly, I recommend performing *in vivo* one-step PET degradation using this system to maximize the advantages of the whole-cell biocatalyst. This attempt would make this study more competitive to reach industrial scale-up by suggesting the possibility to assimilate PET and recycle PET to other valuable products with greatly high efficiency using cell factories.

Thanks for the reviewer's suggestion. At present, our reaction system is established *in vitro*, which will indeed affect the application of industrial massive production. Therefore, by referring to the whole cell catalysis method (Mar. Microb Biotechnol., 2021), we tried to use the displayed cells hydrolyze PET during the growth of medium to achieve PET degradation *in vivo* to maximize the advantages of whole-cell biocatalysts. We first induced the displayed cells in BMM medium for 48 h to ensure the expression of PETase and HFBI on the yeast cell surface. To prove that the cells themselves will not degrade the reaction products of PET, we added 20 mM TPA and 20 mM MHET to the cell culture medium, respectively. After 9 days of continuous culture, the supernatant of cell culture was taken for HPLC detection, and the contents of the two products in the cell culture solution were quantified according to the standard curves of TPA and MHET (Figure S5). The results are shown in Figure S26. It can be seen from the experimental results that the amount of TPA and MHET did not change significantly in both the displayed cells and GS115 control cells, indicating that the cells themselves would not degrade TPA and MHET.

Whole cell catalysis experiment.

Subsequently, we performed the whole cell catalytic experiments using hcPET films and lcPET films ($2 \times 0.8 \text{ cm}^2$) as reaction substrates in the culture system after 48 h of induction. The supernatant was taken every 24 h and the contents of MHET and TPA were detected by HPLC. However, no products such as MHET were detected within 9 days. During the experiment, we monitored the cell growth profile and the change of pH value of the medium, as shown in above Figure. The experimental results showed that the pH of the medium decreased to 3 on the third day. Under this pH condition, the turnover rate of the display system is close to 0 (Figure 3b). In addition, according to the cell growth curve, in the case of co-display system, the cells number had reached 7×10^8 on the second day, which was far beyond the conditions where the reaction could occur, and the turnover rate was approximately 0 at this condition (Figure 3c). Based on the above analysis, these factors may be the reason why the whole cell catalytic system cannot degrade PET film.

14. Please check typos/errors throughout the manuscript: many of them including Results and discussion line 120, mimic should be mimic

We apologize for the spelling mistakes in our manuscript. We have revised the spelling and checked the full text.

Reviewer #3 (Remarks to the Author):

This work described very an interesting work about co-displaying hydrophobin and PETase on the surface of *Pichia pastoris* for enhancing degradation of PETase. Firstly, the co-displaying hydrophobin HFBI with PETase on yeast cells surface was confirmed by several technique such as microscopic observation of florescent stained of yeast surface, number of cells adhesion on PET surface, contact angle measurement, MATH. My main concern of this work is for saying that co-display cells exhibit much higher turnover rates than those of native PETase or PETase displaying cells. The turnover rate was calculated by normalizing the mole of MHET produced per degradation time to the moles of PETase present in the experiments. Protein concentration of PETase displayed was normalized for the comparison of turnover rate. Therefore, the accurate determination of the amount of PETase displayed on yeast surface is very important for calculating the turnover rate. The amount of displayed PETase was determined by Western blot using a gray-scale calibration curve as shown in Figure S9 (a). The question is how can you sure that displayed PETase on yeast surface was completely removed from surface for SDS-PAGE Western blot analysis. In the material and method section, there are no description about how the co-displayed or displayed PETase was removed from yeast surface for Western blot analysis. If less amount of displayed PETase was determined, then it possible leads to a higher turnover number.

Thanks for the reviewer's nice comments. To quantify the displayed PETase concentration, we used Western blot for quantitative analysis. First, we used PETase for Western blot grayscale analysis and plotted a standard curve, as shown in Figure S6a-b. Subsequently, we took 2×10^7 yeast cells after 48 h induction for SDS-PAGE electrophoresis and quantitative analysis by Western blot, as shown in Fig. S6c. According to the standard curve, for display system, the PETase expression amount of 2×10^7 cells is equivalent to 151.2 ng. And for co-display system, the PETase expression amount of 2×10^7 cells is equivalent to 35.2 ng. PET degradation experiments in the manuscript were carried out by taking corresponding multiples of 2×10^7 yeast cells, and the concentration of PETase was calculated according to the proportion.

Figure S6. Displayed PETase quantitative analysis.

Figure S4. Western blot analysis on whole cells, cell walls and protoplasts of the soluble and non-soluble ratio of PETase expressed.

To make sure that displayed PETase on yeast surface was completely removed from surface for SDS-PAGE Western blot analysis. We used three methods to prepare SDS-PAGE samples: whole cell protein, cell wall protein extraction and protoplast preparation. Detailed experimental methods were as follows:

Before the sample preparation for SDS-PAGE, the OD_{600} value of the yeast solution was measured, and the cell counts were all calculated as the same to 2×10^8 .

For the whole cell proteins, the cells were disrupted in 50 μ L buffer A (20 mM Tris/HCl, pH 7.5, 200 mM NaCl) by Bead beater (SI-D258, Scientific Industrial) at 4 $^{\circ}$ C. Then added 5x SDS-PAGE loading buffer (E153-01, GenStar, China) to the final concentration of 1x. After boil at 100 $^{\circ}$ C for 10 min, samples were centrifuged at 13000 g for 15 min the supernatants were collected and resolved by SDS-PAGE.

For the cell wall proteins, the cells were disrupted in 50 μ L buffer A (20 mM Tris/HCl, pH 7.5, 200 mM NaCl) by Bead beater (SI-D258, Scientific Industrial) at 4 $^{\circ}$ C. Then added 150 μ L 1% (v/v) Triton X-100 to extract cell wall proteins at 4 $^{\circ}$ C for 30 min (*Biotechnol Lett*, 2011; *Genetics*, 1999). Added 5x SDS-PAGE loading buffer (E153-01, GenStar, China) to the final concentration of 1x. After boil at 100 $^{\circ}$ C for 10 min, samples were centrifuged at 13000 g for 15 min the supernatants were collected and resolved by SDS-PAGE.

For protoplast proteins, the cells were resuspended by 600 μ L sorbitol buffer, and added 25

U lyticase (Tiangen), then incubated at 4 °C for 30 min. Samples were centrifuged at 1500 g for 10 min and the cell pellets were collected (*Enzyme Microb Technol.*, 2020). Subsequently, the cell pellets were disrupted in 50 μ L buffer A (20 mM Tris/HCl, pH 7.5, 200 mM NaCl) by Bead beater (SI-D258, Scientific Industrial) at 4 °C. Then added 5x SDS-PAGE loading buffer (E153-01, GenStar, China) to the final concentration of 1x. After boiling at 100 °C for 10 min, samples were centrifuged at 13000 g for 15 min the supernatants were collected and resolved by SDS-PAGE.

Then the samples were analyzed by SDS-PAGE and Western blot, and the results were shown in Figure S4. For Co-display system, compared with the expression of PETase in the whole cell, the protein expression on the cell wall accounted for $101 \pm 4.31\%$, while the protoplast samples had almost no protein. The results of the Display system were similar to those of the Co-display system, which indicated that PETase were all displayed on the cell surface, and the degradation of PET was not in the form of free enzymes. Our experimental method can ensure that all the co-displayed or displayed PETase was removed from yeast surface for Western blot analysis. Thus ensuring the accuracy of turnover rate calculation.

The optimization of the degradation system Fig. 3 a is better also compared with PETase-displayed cells and the PETase concentration per cell should be included for comparison. For example 10 displayed PETase per cell compared with 1 per cell, based on the same amount of PETase for turnover, the case with 1 displayed PETase per cell should be more efficient for degradation because the 1 per cell case can have about 10 spots for degradation on PET surface.

Thanks for the reviewer's suggestion. We have added a new Figure S17 to compare the display system and the co-display system for optimizing the degradation system. It can be seen from the experimental results that the turnover rate of hcPET in the display system was significantly lower than that in the co-display system, and the optimal reaction temperature and pH value were the same as those in the co-display system. The turnover rate of the co-display system to hcPET is 9.42 times that of display system. In addition, we compared the amount of PETase on each cell of the co-display system and the display system based on the results of Western blot, as shown in the following table. It can be seen that the amount of PETase on each cell of the display system was 4.3 times more than that of the co-display cells. Therefore, as proposed by the reviewer, the less PETase protein on each cell, the higher the turnover rate. The hydrophobin in the co-display system plays a key role in improving the adsorption of PET by yeast cells.

	PETase concentration per cell ($\times 10^{-6}$ ng)	Turnover rate ($\times 10^{-5}$ sec $^{-1}$)
Co-display	1.76	2146.5 ± 59.5
Display	7.58	227.9 ± 7.68

Figure S17. Comparison of display and co-display systems.

Besides, in addition to the enhanced adhesion to PET surface that leads to the enhanced turnover rate. The mechanism for degradation a PET solid surface by a catalytic microparticles (PETase-displayed yeast cells) should be proposed to explain the enhanced adhesion can promote the degradation.

Thanks for the reviewer's suggestion. To explain the enhanced adhesion can promote the PET degradation, and in order to simulate the two-step process of PET degradation in the co-display system, we performed molecular dynamics simulation analysis. Molecular dynamics simulation seems to be a relatively reliable solution at present. This method has been adopted by several researchers to investigate the interaction of wild-type and mutant PETase and its substrate PET in free state (*ACS Catal.* 2022; *ACS Catal.* 2021; *Proc Natl Acad Sci USA*, 2018; *Proteins*, 2021; *J. Chem. Inf. Model.*, 2021; *ACS Sustain Chem Eng.*, 2021).

Figure 5. The co-display system degrades PET in two steps.

We performed molecular dynamics simulation analysis (Figure 5, Figure S26 and Movie 1 and 2). In the pre-equilibrated system (0 ns), randomly distributed 4PET did not interact with the active center of PETase nor hydrophobic region of HFBI (Fig. 5a and Fig. S26). As time went on, 4PET gradually aggregated near the hydrophobic amino acids of HFBI at about 40-50 ns. At about 60-70 ns, 4PET completely attached to the hydrophobic patch of HFBI mainly through interaction of hydrophobic residues and remain mostly bound for the rest of the trajectory. During this period, 4PET began to gather near the active center of PETase, eventually a number of 4PET molecules were gathered at about 100 ns. The calculation of the distances between 4PET and the other two proteins collaborated with the possible role of HFBI in local enrichment of PETs (Figure S27). From the simulated trajectories, we postulated that our co-display system complete PET degradation through two-step processes of adsorption and hydrolysis, wherein the hydrophobin HFBI favorably binds to the PET chains and grabs them firmly (Fig. 5b), and then PETase interacts with PET and exerts hydrolysis function (Fig. 5c).

Next, in order to obtain a more accurate interaction between PETase-linker-GCW51 and hcPET, and to elucidate whether PETase-linker-GCW51 follows a conformational selection or induced-fit mechanism, we performed an induced fit molecular docking analysis of PETase-linker-GCW51 with a PET tetramer representing the polymer substrate (4PET). The five highest-scoring docking poses were subjected to analysis with respect to spatial arrangement of the residues of the catalytic triad and the conformation of the reactive part of the oligomer substrate (*ACS Catal.*, 2022). And we found the best-predicted docking pose taking

productivity of the catalytic triad into account is shown in Figure 5c. The reacting carbonyl carbon of the substrate was bound with the scissile ethylene glycol moiety in the canonical gauche conformation (Ψ_{gauche}) with the chain twisted. This was consistent with the way in which wild-type PETase binds to 4PET (*ACS Catal.*, 2022).

Figure S28. Schematic diagram of hydrolysis of hcPET by Co-display system.

In order to more clearly understand the two-step process of hcPET degradation by the co-display system, we mapped the mechanism as shown in Figure S28. Firstly, due to the presence of HFBI, Co-display cells quickly adsorbed to the hcPET surface, and the adsorption rate on the hcPET surface was close to 100% (Figure S10 and S11). This could also be further confirmed by the results of scanning electron microscopy. Whether it was lcPET or hcPET, the corrosion spots observed on the surface basically cover the surface, while wild-type PETase only cuts in part of the PET area (Figure 3, S15, S20). Secondly, PETase contacts the surface of high crystallinity PET, and then hydrolyzes the PET chains, thus achieving the effect of efficient hydrolysis of high crystallinity PET.

REVIEWER COMMENTS

Reviewer #2 (Remarks to the Author):

Authors performed additional experiments and more thorough analyses according to my comments. The manuscript is much improved. I do not have any further comment.

Reviewer #3 (Remarks to the Author):

After thorough revision, this manuscript could be accepted for publication,. The authors have replied most of the reviewer's comments.

Reviewer #4 (Remarks to the Author):

The authors use a co-display strategy to functionalize yeast cells with hydrophobin and PETase. The authors show that these cells have enhanced adherence to PET substrates and enhanced degradation of PET compared to PETase alone. Sufficient controls are in place to support the conclusions of this study. This work clearly demonstrates how a whole-cell biocatalyst could be used to degrade PET. This work is timely because PET is one of the most-used plastics and recycling plastic through traditional means is not always cost-effective, meaning that lots of plastic ends up in the environment or landfills. This manuscript is suitable for publication in nature communications, although there were numerous grammatical errors (noted below) that should be improved.

Major points:

-The degradation rate is given as 10.9% in 30 days. It is hard to know what 10.9% means. Is this 10.9% of a gram or 100 grams of PET. Providing the measurement as grams/30 days/ amount of enzyme would make it easier to compare with other past and future studies. A similar comment applies to the rates noted in line 342. Maybe this is not a true reaction rate and if so these results should be reworded.

Minor points:

-In the introduction HFB1 is not defined or introduced before the term is used. Also it is unclear why the hydrophobin SC3 is discussed when it is not used in the manuscript.
-line 139: "performed molecular dynamics"
-line 152: "to response the detecting of anti-flag antibody"
-line 158: "apart from the who cell fraction"
-line 165: "we eager to know whether"
-line 348: "condition, and long term enzymatic"
-line 375: "after lyophilization, which is a dehydration"
-line 377: "after freeze-drying, suggesting"
-Control experiments lacking HFB1 (lines 248-57) seem to be discussed out of place and may belong later in the manuscript.

Point-to-point Response to Reviewers' Comments

We thank all reviewers for their very positive comments and are happy to get the chance to address the errors and suggestions that they have pointed out.

Reviewer #2 (Remarks to the Author):

Authors performed additional experiments and more thorough analyses according to my comments.

The manuscript is much improved. I do not have any further comment.

We thank the reviewer for the appreciation of our work, your encouraging comment is greatly appreciated.

Reviewer #3 (Remarks to the Author):

After thorough revision, this manuscript could be accepted for publication,. The authors have replied most of the reviewer's comments.

We thank the reviewer for careful reading of our manuscript. Thank you very much!

Reviewer #4 (Remarks to the Author):

The authors use a co-display strategy to functionalize yeast cells with hydrophobin and PETase. The authors show that these cells have enhanced adherence to PET substrates and enhanced degradation of PET compared to PETase alone. Sufficient controls are in place to support the conclusions of this study. This work clearly demonstrates how a whole-cell biocatalyst could be used to degrade PET. This work is timely because PET is one of the most-used plastics and recycling plastic through traditional means is not always cost-effective, meaning that lots of plastic ends up in the environment or landfills. This manuscript is suitable for publication in nature communications, although there were numerous grammatical errors (noted below) that should be improved.

We thank the reviewer for the appreciation of our work, your encouraging comment is greatly appreciated. We have carefully revised the manuscript in response to your comments. And we have used Springer Nature Author Services to modify the manuscript.

Major points:

-The degradation rate is given as 10.9% in 30 days. It is hard to know what 10.9% means. Is this 10.9% of a gram or 100 grams of PET. Providing the measurement as grams/30 days/ amount of enzyme would make it easier to compare with other past and future studies. A similar comment applies to the rates noted in line 342. Maybe this is not a true reaction rate and if so these results should be reworded.

Thanks for the reviewer's helpful comment. We have calculated the conversion level in the way you mentioned, and made the following changes in the manuscript:

The resultant **conversion level** for hcPET was increased to **approximately 10.9% (depolymerization rate of 20.92 mg_{products} d⁻¹ mg_{enzyme}⁻¹)** compared with **that (0.003%)** of native PETase within 10 days. **(Lines 32-34).**

The corresponding **conversion level** for hcPET **was approximately 10.9% (depolymerization rate of 20.92 mg_{products} d⁻¹ mg_{enzyme}⁻¹)** at 30 °C within 10 days (Tournier, V. et al. *Nature* 580, 2020, 216-219). **(Lines 110-111).**

The **conversion levels** of hcPET and lcPET within 18 h at 30 °C were **approximately 3.0% (depolymerization rate of 3.27 mg_{products} h⁻¹ mg_{enzyme}⁻¹)** and **55% (depolymerization rate of 178.15 mg_{products} h⁻¹ mg_{enzyme}⁻¹)**, respectively. **(Lines 347-349).**

In addition, we have supplemented the calculation method in the Methods section of the manuscript (Lines 654-662):

The specific activity (SA)¹³ of the enzyme during the PET depolymerization reaction, in mg of equivalent products generated per h per mg of enzyme (mg_{products} h⁻¹ mg_{enzyme}⁻¹) for 18 h reactions (1), and in mg of equivalent total products generated per d per mg of enzyme (mg_{products} d⁻¹ mg_{enzyme}⁻¹) for prolonged reactions (2), was determined by monitoring the liberation of the total products of terephthalic acid (TPA), mono(ethylene terephthalate) (MHET) and bis(2-hydroxyethyl) terephthalate (BHET). TPA, MHET and BHET were measured according to standard curves drawn below, as prepared from commercial TPA and BHET.

$$SA_1 = \frac{\Delta m_{PET}}{18 \times m_{PETase}} \quad (1)$$

$$SA_2 = \frac{\Delta m_{PET}}{10 \times m_{PETase}} \quad (2)$$

Minor points:

-In the introduction HFBI is not defined or introduced before the term is used. Also it is unclear why the hydrophobin SC3 is discussed when it is not used in the manuscript.

Thanks for the reviewer's helpful comment. We have added a description of HFBI in the manuscript and cited the corresponding references (Lines 103-105):

We used the class II hydrophobin HFBI from *Trichoderma reesei* as an example, which is a small, amphiphilic globular protein that readily self-assembles at hydrophilic and hydrophobic interfaces^{54,55}.

In addition, hydrophobin SC3 was discussed here to illustrate that hydrophobins can mediate fungal attachment to the hydrophobic surfaces. It shows that hydrophobin can be used for cell immobilization.

-line 139: "performed molecular dynamics"

-line 152: "to response the detecting of anti-flag antibody"

-line 158: "apart from the who cell fraction"

-line 165: "we eager to know whether"

-line 348: "condition, and long term enzymatic"

-line 375: "after lyophilization, which is a dehydration"

-line 377: "after freeze-drying, suggesting"

We apologize for those mistakes. We carefully checked the content of the text and made changes. In addition, we have used Springer Nature Author Services to modify the manuscript.

-Control experiments lacking HFBI (lines 248-57) seem to be discussed out of place and may belong later in the manuscript.

Thanks for the helpful comment. Here we discussed the control experiments lacking HFBI, mainly to prove that hydrophobin HFBI was important for the adsorption of the co-display system. And these experiments confirmed that hydrophobin HFBI displayed on the yeast cell surface introduced the extra binding capacity to the yeast cell on the PET surface. Therefore, this part was discussed in the adsorption part of the co-display system rather than in the later of the manuscript.

REVIEWERS' COMMENTS

Reviewer #4 (Remarks to the Author):

The reviewers have addressed all of my comments regarding the article. I do not have any concerns.